# Hi3DEval: Advancing 3D Generation Evaluation with Hierarchical Validity

**Yuhan Zhang**[1,2*]   **Long Zhuo**[2*]   **Ziyang Chu**[2,3*]   **Tong Wu**[4†]   **Zhibing Li**[2,5]
**Liang Pan**[2†]   **Dahua Lin**[2,5]   **Ziwei Liu**[6†]

[1]Fudan University   [2]Shanghai Artificial Intelligence Laboratory
[3]Tsinghua University   [4]Stanford University   [5]The Chinese University of Hong Kong
[6]S-Lab, Nanyang Technological University

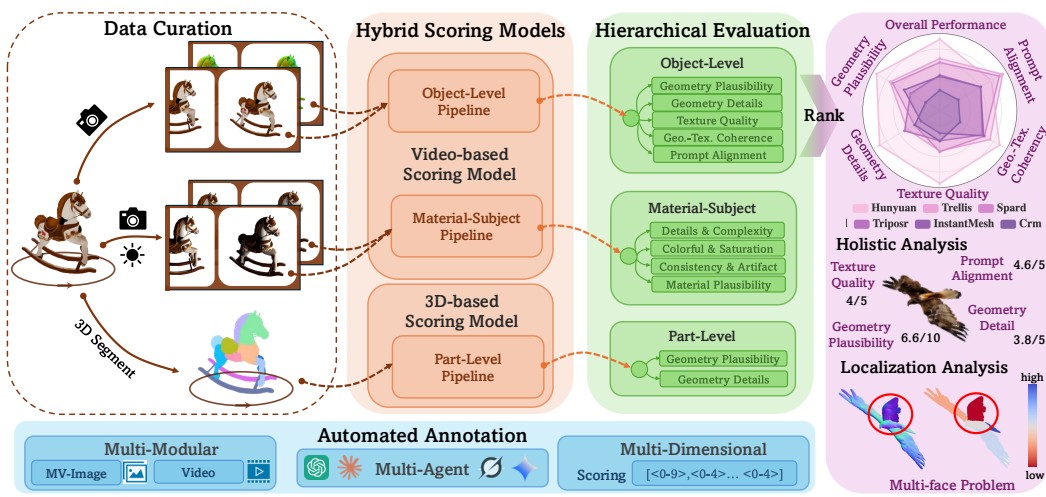

Figure 1: **Overview of Hi3DEval**, a unified framework for 3D generation evaluation with three key components: **1) Hierarchical evaluation protocols** that jointly assess object-level and part-level quality, with extended material evaluation via reflectance cues. **2) A large-scale benchmark** featuring a diverse set of 3D generative models and extensive human-aligned annotations generated via a multi-agent, multi-modal LLMs pipeline. **3) A hybrid automated scoring system** that integrates video-based and naive 3D-based representations to enhance evaluators' perceptions of 3D structure.

## Abstract

Despite rapid advances in 3D content generation, quality assessment for the generated 3D assets remains challenging. Existing methods mainly rely on image-based metrics and operate solely at the object level, limiting their ability to capture spatial coherence, material authenticity, and high-fidelity local details. **1)** To address these challenges, we introduce **Hi3DEval**, a hierarchical evaluation framework tailored for 3D generative content. It combines both object-level and part-level evaluation, enabling holistic assessments across multiple dimensions as well as fine-grained quality analysis. Additionally, we extend texture evaluation beyond aesthetic appearance by explicitly assessing material realism, focusing on attributes such as albedo, saturation, and metallicness. **2)** To support this framework, we construct **Hi3DBench**, a large-scale dataset comprising diverse 3D assets and high-quality annotations, accompanied by a reliable multi-agent annotation pipeline. We further propose a 3D-aware automated scoring system based on hybrid 3D representations. Specifically, we leverage video-based representations for object-level and

material-subject evaluations to enhance modeling of spatio-temporal consistency and employ pretrained 3D features for part-level perception. Extensive experiments demonstrate that our approach outperforms existing image-based metrics in modeling 3D characteristics and achieves superior alignment with human preference, providing a scalable alternative to manual evaluations.

# 1 Introduction

Creating vivid and high-fidelity 3D assets remains a fundamental yet challenging problem in computer vision and graphics, with wide-ranging applications in gaming, virtual and augmented reality, and robotics. In recent years, the field has witnessed significant breakthroughs, driven by the emergence of large-scale datasets [8, 7], expressive neural representations [37, 23], sophisticated optimization techniques [40, 45, 28, 29, 4], and powerful network architectures [63, 72, 74, 52]. As the visual realism of generated content continues to advance, establishing reliable and fine-grained metrics for evaluating and comparing different approaches has become a critical objective.

Existing 3D evaluation frameworks can be broadly categorized into two paradigms: training-free protocols and data-driven evaluators. The former often extend conventional 2D metrics to 3D domains through handcrafted heuristics [13, 46], while methods like GPTEval3D [61] leverage powerful Multimodal Language Models (MLLMs) such as GPT-4V [1] with multi-view renderings to enhance 3D reasoning. In pursuit of greater transparency and alignment with human judgment, recent efforts (e.g. 3DGen-Bench [73], Gen3DEval [35], T23DAQA [12]) explore training lightweight scoring models supervised by human annotations or pseudo-labels generated by GPT-4V. Despite these advances, current approaches remain coarse-grained and overlook material attributes when evaluating textures. Moreover, base on 2D renderings, these methods inherently struggle to capture the spatial continuity and structural complexity of 3D assets, limiting their reliability and robustness.

To address these limitations, we present a unified evaluation framework, Hi3DEval, that supports both image- and text-conditional 3D generation and advances 3D evaluation in the following dimensions: **1) Hierarchical evaluation scheme across varying granularities.** We introduce a hierarchical evaluation scheme, supporting both object-level and part-level assessment. Specifically, the object-level evaluation provides a holistic evaluation of generated 3D assets, considering geometry, texture, and prompt alignment. While the part-level evaluation enables localized diagnosis of quality issues within semantic regions, enhancing interpretability and failure analysis. Together, they provide a more comprehensive view of generation performance. **2) Physical material evaluation with reflectance cues.** We propose a material-subject evaluation protocol that goes beyond aesthetic-level judgments to explicitly assess core physical properties, such as albedo, saturation, and metallicness. In practice, we use reflectance cues under diverse illumination to simulate realistic perception scenarios, enabling robust assessment across diverse 3D representations, even when PBR are not explicitly disentangled during generation. **3) Large-scale dataset with human-aligned annotation pipeline.** We construct a large-scale comprehensive dataset involving all aforementioned dimensions, dubbed Hi3DBench. To trade off the subjectivity of purely manual labeling and the inconsistency of purely GPT-based labeling with human judgment, we introduce a multi-agent, multi-modal annotation pipeline, producing more consistent and faithful assessments. **4) Hybrid 3D representation-based automated scoring system.** To overcome the limitations of static image representation, we propose a 3D-aware automated scoring system leveraging hybrid 3D representations. To be specific, we use video-based representations to enhance evaluators' understanding of spatio-temporal consistency for object-level and material-subject evaluation, and we apply pretrained geometric embeddings for the part-level to achieve deep shape perception.

Experimental results show that such hybrid 3D-aware signals yield more reliable and human-consistent evaluation than conventional static image-based approaches. Specifically, the proposed video-based scoring pipeline, adopted for the object-level and material-subject evaluations, achieves superior pairwise alignment with humans across all assessed dimensions. Furthermore, we conduct qualitative studies on the part-level scoring model, demonstrating that it exhibits capability in localizing generation flaws, enabling fine-grained diagnostic analysis.

---

[1] GPT-4V system card: `https://openai.com/index/gpt-4v-system-card`

## 2 Related works

**3D object generation.** Prior work in 3D content generation has explored a variety of approaches, including leveraging 2D generative models, incorporating 3D geometric priors, and directly learning from large-scale 3D datasets. DreamFusion [40] leverage text-to-image diffusion model [17, 43] to optimize differentiable 3D representations through Score Distillation Sampling (SDS). Subsequent methods [29, 54, 57, 36, 28, 47, 51, 32] have refined optimization framework and SDS loss to enhance visual quality. Another line of research [33, 30, 45, 34] focuses on adapting 2D generative models for multi-view synthesis, providing strong 3D priors that enable reconstruction of 3D assets via sparse-view reconstruction techniques or Large Reconstruction Models [18, 50, 27, 66, 64, 53, 58]. More recently, native 3D generation approaches[72, 77, 63, 74, 52, 5] have achieved state-of-the-art performance by training directly on 3D data collections [8, 7, 62], significantly improving both geometric fidelity and computational efficiency. As generation techniques evolve, the need for an efficient, robust, and comprehensive 3D evaluation framework has become increasingly urgent.

**3D generation evaluation.** Early approaches to evaluating 3D content often relied on labor-intensive user studies or 2D-based metrics such as CLIP Score [15, 20] and Aesthetic Score [44], which are insufficient for capturing the holistic structure of 3D assets. To address these limitations, $T^3$Bench [13] proposed aggregating multi-view image features via hand-crafted formulations. With the growing adoption of the "LLM-as-a-Judge" [76] paradigm, GPTEval3D [61] utilized GPT-4V to perform pairwise comparisons and established a leaderboard using Elo scoring [10]. However, reliance on a closed-source model raises concerns about reproducibility and transparency. Subsequent efforts [48, 73, 35] fine-tuned open-source models to produce pairwise comparison outcomes using pseudo-labels generated by GPT-4V. Despite progress, this pairwise-to-score pipeline presents scalability challenges as leaderboard sizes increase. As a result, recent works [73, 68, 46, 12] have shifted toward absolute scoring paradigms. For example, GT23D-Bench [46] introduces a training-free protocol in fine-grained dimensions using curated ground-truth sets and conventional metrics, while T23DAQA [12] enhances 3D understanding through video-based representations. Furthermore, driven by the rise of reward modeling (RM), DreamReward [68] and 3DGen-Score [73] learn scoring models from human preference data, achieving stronger alignment with human judgments.

Despite these advancements, two main limitations remain: most methods rely heavily on 2D renderings derived from 3D assets, which fail to capture true 3D structure and spatial consistency, and evaluations are typically constrained to the object level, lacking finer-grained analysis. To address these gaps, our method introduces a hybrid, 3D-aware scoring system and proposes a part-level evaluation framework to enable more detailed and structure-aware assessment.

**Material generation evaluation.** Generating high-quality textures conditioned on 3D geometry [3, 2, 70, 6, 11, 71, 9, 69, 21] has attracted growing interest, highlighting the need for evaluation metrics specifically tailored to this task. However, existing metrics are inadequate for capturing the physical plausibility or alignment of textures with underlying geometry. For example, Frechet Inception Distance (FID) [16] and Kernel Inception Distance (KID) [1] assess the distribution of multi-view renderings, but are not well-suited for object-level evaluation. Similarly, CLIP Score is primarily designed for measuring text-image alignment, while Aesthetic Score offers only a coarse assessment of visual appeal. Crucially, these metrics overlook essential material properties such as albedo, metallic, and roughness, which are fundamental to physical plausibility and perceptual realism. In this work, we incorporate multi-lighting setups and perform fine-grained, physically aware assessments that more accurately reflect the quality and realism of generated textures.

## 3 Hierarchical 3D scoring dataset

To enable systematic evaluation of generative 3D models, we construct **Hi3DBench**, a large-scale benchmark tailored for multi-dimensional and hierarchical quality assessment. In contrast to prior benchmarks [61, 35, 68] that focus solely on object-level pairwise comparisons or limited text-conditioned scenarios, Hi3DBench comprises over 15,000 procedurally generated assets with hierarchical annotations spanning object-, part-, and material-levels in absolute scoring format. Furthermore, to reduce labor-intensive manual annotations and mitigate the subjectivity of human ratings, we introduce a **Multi-agent Multi-modal Annotation Pipeline** (**M²AP**), which harnesses a diverse set of MLLM agents to collaboratively yield scalable, consistent, and reliable quality assessments.

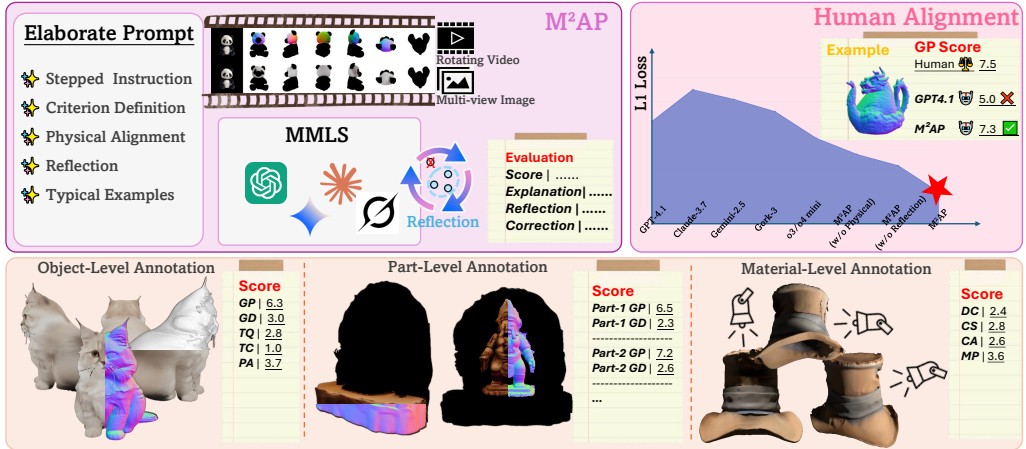

Figure 2: **Annotation of the Hi3DBench Dataset.** The top-left panel illustrates the annotation pipeline of **M²AP**, while the top-right panel presents a human alignment experiment comparing annotations from a single agent and M²AP. The vertical axis denotes the L1 loss between the predicted values and human annotations, with exact numerical results reported in Table R1. As shown, M²AP integrates advanced multimodal agents, incorporates reflection to address hallucination issues, and utilizes an elaborate scoring prompt along with rendered rotation videos and multi-view images to produce the final evaluation. Finally, the bottom row showcases representative annotation examples at the object, part, and material levels.

We detail the dataset construction process in Section 3.1, including asset generation, part segmentation, and relighting; describe the annotation pipeline in Section 3.2 and evaluate its reliability in Section 3.3.

## 3.1 Data curation

To support hierarchical and multi-dimensional evaluation of 3D generative models, we construct Hi3DBench by systematically curating a large-scale collection of procedurally generated 3D assets. The data curation process is designed to ensure diversity, representativity, and compatibility with downstream annotation and evaluation. Specifically, it consists of three key steps: 1) diverse asset generation; 2) part-level segmentation for fine-grained analysis; 3) relighting for material realism.

**Diversified generation.** Our benchmark includes 15,300 assets in total, generated from 30 distinct 3D generative methods (including 9 text-to-3D models and 21 image-to-3D models). For each method, we generated 510 objects with prompts borrowed from 3DGenBench [73], which spanning diverse semantic categories and difficulties. The complete list of models involved and details about this process can be found in Section A.2. Additionally, to facilitate visualization and subsequent evaluation, we render each 3D asset into 360-degree surround videos in three distinct formats: RGB, Normal, and Shading. Meanwhile, to reduce the potential errors introduced by visual quality, we use a unified rendering pipeline and consistent settings to render all methods and assets. More details on the rendering implement can also be found in Section A.2.

**Part-level segmentation.** Following the generation phase, 3D assets undergo a structurally meaningful segmentation process, which is crucial for fine-grained analysis. Specifically, we adopt PartField [31] in a semantics-free manner, which leverages rich local geometric features to perform effective unsupervised clustering. However, the number of part clusters is not automatically inferred by PartField [31] but manually specified instead. Given that structural complexity varies significantly depending on the input prompt, we argue that applying a fixed number of clusters across all objects is suboptimal. For instance, "a potted cactus" can be segmented into three parts (pot, soil, and cactus), whereas "a kitty cat"" exhibits richer structural details (e.g., head, torso, limbs and tail) that demands a finer partition. To accommodate such variation, we utilize GPT to estimate an appropriate number of structurally meaningful parts for each prompt based on its semantic complexity, followed by manual validation to ensure its plausibility and consistency. More details on the selection of the part segmentation method and the impact of the cluster count can be found in Section A.2.

**Relighting.** To ensure a consistent and accurate material assessment across diverse assets, we apply a standardized relighting protocol. Each object is rendered under both controlled point-light conditions and various High Dynamic Range Imaging (HDRI) environments. Specifically, we place point-light

sources at two principal azimuthal angles (top and right) and render from complementary viewpoints to reveal object reflectance characteristics. Further, to simulate real-world scenarios, we adopt six HDRI maps spanning indoor and outdoor environments with varying natural and artificial illumination. This dual-scheme relighting captures material fidelity under both idealized and realistic lighting, enabling comprehensive evaluation. More details about the relighting setting and visualizations of HDRI maps involved are provided in Section A.2.

## 3.2 Annotation pipeline

To enable large-scale, reliable, and cost-efficient quality annotation, we introduce a Multi-agent Multi-modal Annotation Pipeline (M²AP). The pipeline (illustrated in the top-left panel of Figure 2) utilizes with an Elaborate Prompt, incorporating content like Stepped Instructions and Typical Examples to guide the annotation. In practice, M²AP uses Rotating Videos and Multi-view Images as inputs for a comprehensive understanding of 3D assets, and processes rich visual data with advanced image-aware and video-aware Mutimodular LLMs. Additionally, we also employ most-recent advanced reasoning model (i.e., GPT 4.1, Claude 3.7, and Grok-3) and thinking model (i.e., Gemini 2.5 Pro and O3 / O4-mini), dubbed Reflection mechanism, to ensure consistency and alleviate hallucination by self-revision. Further information on the annotation pipeline, including the selected agents, prompt design, and annotation cost, are provided in Section A.3. In the following paragraphs, we demonstrate how it supports hierarchical annotations at object, part, and material levels, respectively.

**Object-level criteria.** At the object level, M²AP provides holistic quality assessments of 3D content across five key dimensions, in line with established 3D evaluation protocols [61, 73].

- *Geometry Plausibility* (GP), assessing the structural integrity and physical feasibility of the generated shape, with absence of distortions or floating parts;
- *Geometry Details* (GD), assessing the fidelity of fine-scale structures, distinguishing well-defined, intentional surface features from noise;
- *Texture Quality* (TQ), assessing the visual fidelity of surface textures in terms of resolution, realism, aesthetics, and consistency across views;
- *Geometry-Texture Coherency* (GTC), assessing the alignment between geometry and texture, ensuring textures naturally follow shape contours and consistently reflect geometric details;
- *Prompt Alignment* (PA), assessing the semantic and/or identity consistency between the input prompt and the generated 3D asset.

**Part-level criteria.** The part-level assessment dives into the segmented components of the 3D object, enabling fine-grained analysis and precise localization of generation flaws. It focuses on *Geometry Plausibility* (GP) and *Geometry Details* (GD) for each part, complementing object-level evaluation by exposing localized artifacts.

**Material-subject criteria.** Material-subject evaluation targets the intrinsic perceptual quality and physical properties of textures, filling a critical gap in prior frameworks with four key dimensions:

- *Details and Complexity* (DC), assessing the texture's visual richness and detail while ensuring a balanced complexity that preserves aesthetic harmony;
- *Colorfulness and Saturation* (CS), assessing the texture's color distribution and clarity, focusing on diversity, saturation, and suitability.
- *Consistency and Artifacts* (CA), assessing the texture's consistency and realism under varying lighting conditions, focusing on the presence of visible seams and shading artifacts;
- *Material Plausibility* (MP), assessing whether its diffuse and specular effects realistically reflect the material properties described in the prompt.

## 3.3 Alignment with human judgments

To assess the reliability of our automated annotation pipeline, we collect a set of human annotations and conduct a human-agent alignment study. As shown in the top-right panel of Figure 2, our proposed M²AP outperforms single-agent baselines by a clear margin in terms of L1 loss, highlighting the advantage of collaborative agent reasoning in producing more consistent and accurate annotations.

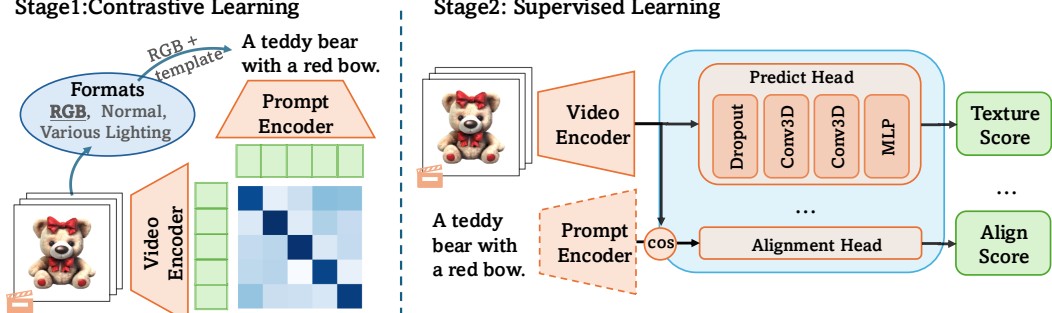

Figure 3: **Overview of video-based scoring pipeline. Left**: Contrastive learning aligns the video encoder with the prompt encoder under diverse rendering conditions. **Right**: Quality heads are trained to regress scores. Specifically, we apply cosine similarity for prompt-aware dimensions.

In addition, we conduct ablation studies to validate the effectiveness of physical plausibility check and reflection. As shown, the proposed M²AP achieves a lower L1 loss compared to the variants without hallucination mitigation, i.e., via reflection and physical plausibility check. It indicates that M²AP demonstrates strong human alignment. For example, M²AP provided a score of 7.3 for a teapot model, closer to the human score of 7.5 than the standalone GPT-4.1 (5.0). More details on the progress of human annotation and quantitative results of ablation study are provided in Section A.3.

### 3.4 Annotation cost

The proposed M²AP achieves efficient large-scale annotation, requiring on average $0.15 and 20–60 seconds per object in a non-parallelized setup. Overall, the pipeline produced 4K object-level, 23K part-level, and 11K material-subject annotations, with a total cost of approximately $4.1K. A breakdown of API expenses is provided in Table R2..

## 4 Hybrid 3D-aware scoring system

### 4.1 Video-based scoring model

To robustly assess the quality of 3D assets, we adopt a video-based evaluation paradigm that captures spatio-temporal cues from rendered turntable sequences. This could provide a more comprehensive perception of 3D content compared to image-based representations.

**Model.** We leverage the pretrained InternVideo2.5 [56] encoder, which demonstrates effective video-text alignment in embedding space, to extract rich spatiotemporal features from multi-modal rendered videos. Following common practices in video quality assessment [59], we design a lightweight prediction head to process the high-dimensional features from the video encoder. The unit structure of the prediction head comprises two Conv3D layers with GeLU activations to model spatiotemporal dependencies across frames. Finally, a Linear layer outputs the scalar quality score. This setup leverages the strong spatio-temporal representations of the video encoder while maintaining efficiency and generalizes well across object categories, materials, and lighting variations.

**Pipeline.** While the video encoder demonstrates strong video-text alignment, we observe obvious domain gaps when processing 3D rendered videos. To address this, we implement a two-stage training strategy, as illustrated in Figure 3. In the first stage, we curate a diverse dataset including both scanned and generated 3D objects and render them under various visual conditions (e.g., albedo, normal, lighting). Through contrastive learning, we align the rendered videos with their corresponding prompts and conditioning descriptions, guided by CLIP [42] pretrained encoders as the target embedding space. In the second stage, we train the quality prediction head together with the final two MLP layers of the video encoder, using human-aligned quality scores. This enables domain-specific adaptation for scoring while preserving the general spatio-temporal representations.

**Criterion.** Since annotations are collaboratively provided by multiple agents, the resulting scores lie on a continuous scale. Therefore, we treat the scoring task as a regression problem and utilize SmoothL1 loss as the primary objective, which combines the stability for small errors of L2 loss with the robustness for outliers of L1 loss. Furthermore, to capture relative quality judgments and enhance

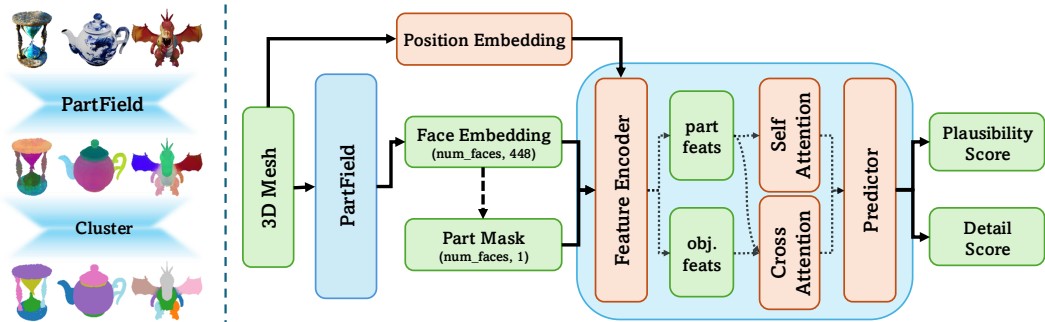

Figure 4: **Overview of the 3D-based scoring pipeline. Left**: Visualizations of the raw mesh, face embeddings, and part masks used in the pipeline, from top to bottom, exhibiting the strong capability of PartField [31] in capturing local geometric features. **Right**: Illustration of the scoring pipeline. We first project the pretrained features into a latent space tailored for the scoring task, then apply attention modules to enable information flow with the global context and within each part.

the model's discriminative ability, we also incorporate ranking loss as an auxiliary objective. The final loss function is formulated as follows.

$$\mathcal{L}_{\text{reg}} = \begin{cases} 0.5(s - \hat{s})^2/bata, & \text{if } |s - \hat{s}| < bata, \\ |s - \hat{s}| - 0.5 * bata, & \text{otherwise,} \end{cases} \tag{1}$$

$$\mathcal{L}_{\text{rank}} = \sum_{i,j} \max(0, -(s_i - s_j)(\hat{s}_i - \hat{s}_j)), \tag{2}$$

$$\mathcal{L}_{\text{total}} = \mathcal{L}_{\text{reg}} + \lambda \mathcal{L}_{\text{rank}}, \tag{3}$$

where $s$ denotes the ground-truth score and $\hat{s}$ denotes the predicted score, and the constant $\lambda$ controls the strength of the ranking supervision. In this paper, we set the default value $bata$ at 1.0.

**Object-level setup.** In the first stage, we choose 6k scanned objects from Omniobjs [62] and 6k generated objects, which are rendered to 16-frame videos of albedo, normal, point-light, and HDRI types. The video encoder is fully fine-tuned on video-prompt pairs with contrastive loss. We use 4 NVIDIA A800-SXM4-80GB GPUs and train the encoder for approximately 4 hours. During the second stage, five prediction heads are trained in parallel using around 10k samples per dimension, with each head corresponding to the specific video types (normal, RGB).

**Material-subject setup.** The first stage is identical to Object-level Setup. For the second stage, we independently train the four parallel prediction heads using approximately 10k samples per dimension. The trainable parameters are about 37M per dimension using an Adam optimizer with a learning rate of 4e-4. The batch size is set to 16 per GPU with 16 frames per video. Each dimension completes training on 8 NVIDIA A800-SXM4-80GB GPUs in about 8 hours for 15 epochs.

### 4.2 3D-based scoring model

**Preliminary.** PartField [31] is a feedforward class-agnostic part-segmentation model that learns a continuous 3D feature field for part-aware shape understanding. It is trained using weak part proposals derived from 2D segmentations and 3D part annotations, without enforcing consistent semantics or granularity. This flexible supervision enables cross-modality generalization (e.g., meshes, point clouds, and Gaussian splats) and robust representations under open-world.

**Pipeline.** We build our part-level scoring pipeline on top of PartField [31], which provides strong local geometric features from unsupervised 3D segmentation. As shown in Figure 4, starting from these pretrained features, we project them into a scoring-specific latent space via a two-layer encoder. Additionally, to handle the varying number of mesh faces per part, we apply the adjacency-based k-NN pooling to standardize into a fixed-dimensional representation. Operating on local structure solely, they lack broader contextual understanding. To address this, we introduce two complementary interaction modules: a cross-attention module that allows incorporation of global contextual cues and a self-attention module that captures intra-part dependencies. This design enables the model to capture both fine-grained geometric details and holistic structural context for precise assessment.

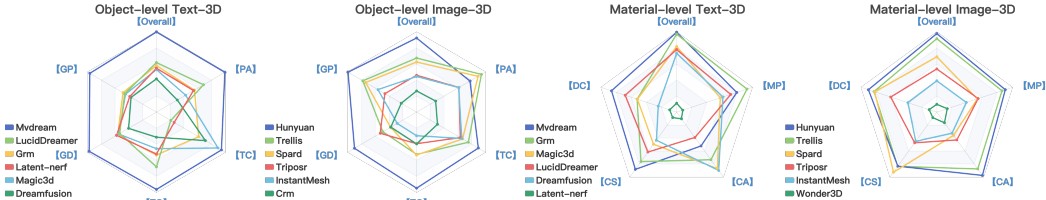

Figure 5: **Radar visualization of Hi3DBench.** Models are ranked top-down by their total score across all dimensions, each predicted by our automated scoring system. The two charts on the left show object-level results, while the two right show material-subject results. For clarity, only the top 6 models are shown in the charts. And the full leaderboards are provided in Section B.3.

Table 1: **Pairwise rating alignment at the object level.** We report the average pairwise accuracy across different scoring models. Note that ImageReward [65] and GPTEval3D [61] are tailored specifically for text-to-3D generation, and thus entries for image-to-3D are left blank.

| Metrics | Text-to-3D | | | | | Image-to-3D | | | | |
|---|---|---|---|---|---|---|---|---|---|---|
| | GP | GD | TQ | GTC | PA | GP | GD | TQ | GTC | PA |
| CLIP Score [42] | 0.556 | 0.580 | 0.606 | 0.556 | 0.604 | 0.589 | 0.588 | 0.605 | 0.636 | 0.623 |
| ViCLIP Score [55] | 0.557 | 0.591 | 0.625 | 0.577 | 0.617 | 0.589 | 0.570 | 0.611 | 0.640 | 0.623 |
| Aesthetic Score [44] | 0.657 | 0.634 | 0.607 | 0.629 | 0.623 | 0.570 | 0.613 | 0.622 | 0.675 | 0.630 |
| Image Reward [65] | 0.568 | 0.598 | 0.607 | 0.513 | 0.610 | - | - | - | - | - |
| GPTEval3D [61] | 0.690 | 0.689 | 0.677 | 0.667 | 0.649 | - | - | - | - | - |
| **Ours** | **0.774** | **0.725** | **0.755** | **0.749** | **0.726** | **0.718** | **0.703** | **0.753** | **0.732** | **0.710** |

**Part-level setup.** Our scoring model contains approximately 5 million trainable parameters and is trained on 22.7k samples per evaluation dimension. Prior to training, we normalize all ground-truth scores into the [0,1] range and apply a weighted sampling strategy to balance the original score distribution in the training set. Moreover, to encourage the intra-object part-level information exchange, we adopt the object as the minimal input unit instead of isolated parts. Specifically, each training batch consists of 8 objects, averaging around 6 parts per object. The model is trained for a total of 5000 steps using the Adam optimizer with a learning rate of 3e-5. Training is performed on a single NVIDIA A100-SXM4-80GB GPU and is completed in approximately 4 hours.

## 5 Evaluation

To systematically assess generative performance, we employ our automated scoring system to construct comprehensive leaderboards, as illustrated in Figure 5. These leaderboards offer an objective and reproducible benchmark of state-of-the-art methods across multiple evaluation dimensions. Beyond model ranking, we further evaluate the reliability and robustness of our scoring system through extensive qualitative and quantitative analyses. As detailed in Section 5.1 and Section 5.2, these evaluations are organized hierarchically, spanning from coarse-grained object-level assessment to finer-grained material-subject and part-level evaluations, thereby demonstrating the effectiveness of our system across varying levels of detail. Additionally, we conduct ablation studies to systematically validate the design choices and contributions of individual components within the scoring pipeline. Detailed experimental settings and results are presented in Section B.1 and Section C.

### 5.1 Quantitative analysis

**Baseline metrics.** In our evaluation, we incorporate 5 representative baseline metrics, encompassing CLIPScore [15], ViCLIP [55], Aesthetic score [44], ImageReward [65] and GPTEval3D [61]. Each metric is computed on our test set and averaged on 40 views. For GPTEval3D [61], we replace the original GPT-4v model with the more advanced GPT-4o when calculating pairwise comparisons. Additionally, noting that GPTEval3D [61] and ImageReward [65] are designed exclusively for text-to-3D evaluation, therefore they are not applicable to image-to-3D models.

**Object-level evaluation.** In Table 1, we show the pairwise rating alignment in object-level across 1000 test objects pairs sampled from human annotations [73], covering both text-to-3D and image-to-

Table 2: **Pairwise rating alignment at the material level across objects.** We report the average pairwise accuracy across different scoring models. As GPTEval3D [61] does not explicitly account for material properties in its texture assessment, its scores are mapped only to albedo-related dimensions for fair comparison.

| Metrics | Text-to-3D | | | | Image-to-3D | | | |
|---|---|---|---|---|---|---|---|---|
| | DC | CS | CA | MP | DC | CS | CA | MP |
| CLIP Score [42] | 0.647 | 0.607 | 0.543 | 0.640 | 0.699 | 0.678 | 0.604 | 0.689 |
| ViCLIP Score [55] | 0.673 | 0.657 | 0.577 | 0.620 | 0.701 | 0.673 | 0.630 | 0.698 |
| Aesthetic Score [44] | 0.690 | 0.690 | 0.563 | 0.633 | 0.642 | 0.633 | 0.550 | 0.619 |
| Image Reward [65] | 0.627 | 0.613 | 0.540 | 0.613 | - | - | - | - |
| GPTEval3D [61] | 0.630 | 0.614 | - | - | - | - | - | - |
| **Ours** | **0.767** | **0.773** | **0.733** | **0.745** | **0.723** | **0.771** | **0.737** | **0.763** |

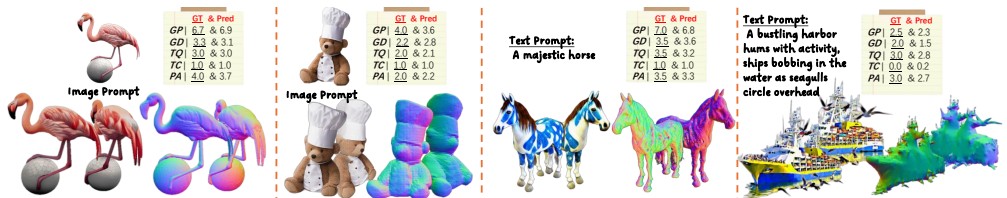

Figure 6: **Visual examples of the object-level scoring model.** We present representative examples in both image-to-3D and text-to-3D settings. For each object, we compare the predicted scores from our model with human ground-truth annotations across all evaluation dimensions. These results demonstrate the model's ability to produce accurate and consistent assessments at the object level.

3D settings. As shown, our model consistently outperforms other evaluation methods in a significant margin and achieves the highest accuracy across all dimensions, highlighting the effectiveness of our scoring system. Additional comparison results with T3Bench [13] are provided in Section B.2.

**Material-subject evaluation.** For material evaluation, we sample 1000 image-to-3D pairs and 300 text-to-3D pairs from the test set to comprehensively assess the performance of our scoring model. As reported in Table 2, our model demonstrates strong alignment with human judgments, comprehensively outperforming baseline metrics across all dimensions. Notably, our model excels in lighting-sensitive dimensions such as *Consistency and Artifact*, indicating its capability to capture subtle material properties affected by illumination and artifacts.

## 5.2 Qualitative analysis

**Object-level evaluation.** We illustrate the scoring performance at the object level in Figure 6. As shown, our scoring system is capable of producing accurate and consistent scores for both text-to-3D and image-to-3D objects across all dimensions. This demonstrates that our scoring model could capture and analyze 3D-aware visual clues, such as geometric structure and texture fidelity, highlighting its capability to perform holistic assessments at the object level.

**Part-level evaluation.** The part-level evaluation paradigm enables a more nuanced understanding of geometric quality within 3D assets by decomposing the object into semantically coherent components. This granularity allows the model to isolate and assess localized imperfections, such as collapses, distortions, or multi-faces, that are often obscured in holistic object-level evaluations. As shown in Figure 7, for *Geometry Plausibility*, this fine-grained perspective facilitates the detection of subregions that critically undermine overall perceptual quality, such as body distortions in the frog or extraneous limbs in the cat. For *Geometry Detail*, it provides a comprehensive view of the spatial distribution of fine-scale features, distinguishing meaningful details from noise.

**Material-level evaluation.** Our material-level evaluation focuses on more detailed analysis of texture. As illustrated in Figure 8, our model is able to look for differences in both details and lighting reactions and examine whether the texture maintains a suitable visual harmony. For *Consistency and Artifacts*, our model is capable of handling various conditions of lights and capturing critical defects

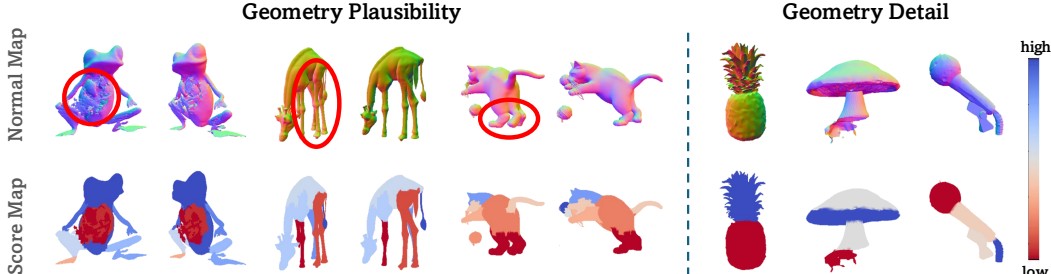

Figure 7: **Visual examples of the part-level scoring model.** We apply a normalized colormap to visualize part-level scores within objects, where blue indicates high-quality regions and red denotes low-quality regions. **Left**: Our scoring can locate surface distortions and abnormal structures in terms of geometry plausibility. **Right**: Our scoring can reflect the spatial distribution of geometric details.

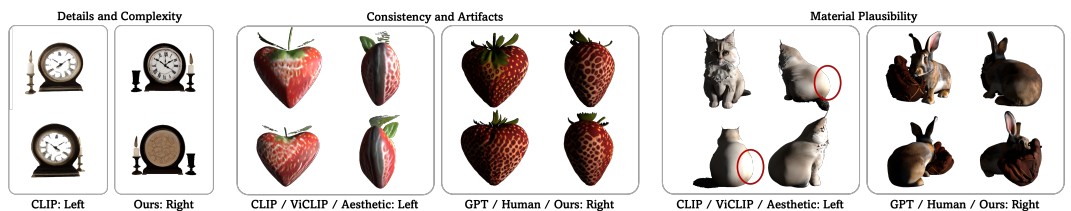

Figure 8: **Visual examples of the material-level scoring model.** We compare our score rating results with baseline metrics and human annotations. Our model can accurately capture the texture representations for detail analysis, observe obvious lighting differences or shading artifacts for lighting conditions, and understand basic reflection rules for material plausibility.

when turning to side or back. For *Material Plausibility*, our model can understand the object without additional prompt information and judge the correctness of reflections.

## 6  Conclusion

This work takes a significant step toward systematic evaluation of conditional 3D generation by proposing a unified framework that integrates hierarchical granularity, physical material realism, and automated annotation. By combining object-level and part-level assessment, our framework enables both holistic assessments and fine-grained quality diagnosis. The introduction of a multi-agent annotation pipeline further enables scalable and human-aligned dataset construction, supporting robust model training and evaluation. Our hybrid 3D-aware scoring models, grounded in both geometry and video-based representations, offer a promising alternative to traditional image-based proxies. We believe this framework can facilitate broader efforts toward generalizable 3D quality assessment, and serve as a viable alternative to human evaluation.

**Limitations and future work.** Although our framework advances hierarchical evaluation for 3D object generation, it currently focuses on object-centric, static assets. Extending the evaluation paradigm to compositional scenes or dynamic content remains an open challenge for future work. Second, our part-level annotation framework requires high-quality mesh segmentation and assumes consistent part semantics across different object instances, which may be unreliable for highly deformable or abstract shapes. Future work will aim to extend the evaluation paradigm to more complex scenarios, including dynamic and scene-level compositions, and explore adaptive segmentation strategies and more sophisticated multi-modal integration methods, thereby improving generalization.

**Societal Impacts.** While the automated evaluation pipeline enhances scalability and consistency, it may inadvertently reinforce biases present in training data or propagate subjective quality norms at scale. We encourage responsible use of our framework and ethical deployment in applications.

## Acknowledgements

This study is supported by Shanghai Artificial Intelligence Laboratory and the Ministry of Education, Singapore, under its MOE AcRF Tier 2 (MOET2EP20221-0012, MOE-T2EP20223-0002). This research is also supported by cash and in-kind funding from NTU S-Lab and industry partner(s).

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

# Appendix

## A More details about scoring dataset

### A.1 Overview

In this paper, we conduct a comprehensive 3D evaluation dataset with hierarchical annotations based on 3DGen-Bench [73]. To be specific, we extend the object-level annotations in 3DGen-Bench [73], and propose novel part-level and material-subject evaluation through customized preprocessing (e.g., part segmentation and relighting). In summary, we obtain 15.3k synthesized 3D assets, with 4k related object-level annotations, 23k part-level annotations, and 11k material-subject annotations. The complete dataset can be accessed and downloaded publicly from the Huggingface repo: `https://huggingface.co/datasets/3DTopia/Hi3DBench`.

### A.2 Data curation

**3D generative models involved.** Our benchmark includes 30 3D generative models in total, including 9 text-to-3D models and 21 image-to-3D models. The full list of involved models is provided below.

- **9 Text-to-3D models**: MVDream [45], LucidDreamer [28], GRM [66], Magic3D [29], Latent-NeRF [36], DreamFusion [40], SJC [54], Point-E [38], Shap-E [22].
- **21 Image-to-3D models**: Hunyuan3D 2.0 [74], Trellis [63], SPAR3D [19], TripoSR [52], Unique3D [60], CRM [58], LN3Diff [26], InstantMesh [64], Wonder3D [34], OpenLRM [14], Stable Zero123 [32], Zero-1-to-3 XL [32], Magic123 [41], LGM [50], GRM [66], SyncDreamer [33], Shap-E [22], Triplane-Gaussian [77], Point-E [38], Escher-Net [24], Free3D [75].

**Implementation details.** We generated 510 assets for each method guided by 510 prompts proposed by 3DGen-Bench [73]. All experiments are conducted with the official public code and the default hyperparameters. Notably, Trellis [63] and Hunyuan3D 2.0 [74] only release their code and checkpoints for image-to-3D; thus, we haven't conducted experiments for text prompts. The entire generation process consumes around 2 weeks, utilizing 4 NVIDIA A100-SXM4-80GB GPUs.

**Rendering details.** Our rendering pipeline is implemented using both Blender and Kiui Python tools. Specifically, Blender is used for HDRI and point-light rendering due to its high-quality output, while Kiui is adopted for RGB and normal map rendering, benefiting from its computational efficiency and implementation simplicity. All videos are rendered at a default resolution of 512×512 pixels with 25 FPS, and we maintain consistent lighting and camera parameters across all methods. The average rendering cost per video is 17.47s using Blender.

**Part-level segmentation.** To support part-level annotation and evaluation, we propose to perform part segmentation first. To handle open-vocabulary 3D assets and accommodate structural failures (e.g., mesh collapses or topological inconsistencies) commonly observed in generative outputs, we adopt a semantic-free, category-agnostic partitioning strategy. We evaluate several state-of-the-art methods, such as SAMPart3D [67], SAMesh [49] and PartField [31]. However, SAMPart3D [67] relies on sample-specific optimization and requires approximately 30 minutes per mesh, making it impractical for large-scale preprocessing. Compared to SAMesh [49], lifting 2D segmentation into 3D, PartField [31] builds a hierarchical segmentation tree based on learned feature fields, offering finer control over part granularity and more stable performance. Additionally, its learned features can be directly leveraged in our scoring pipeline. Thus, we adopt PartField [31] as the final codebase. Since the number of parts is not predicted by models, we prompt GPT to assign suitable target part counts for each prompt, as illustrated in Figure S1.

**Relighting.** To accurately capture real-world texture quality and enhance fine-grained detail evaluation, we implement a multi-lighting setup using controlled point lights and environmental HDRI maps. Four point light sources are employed to cover complete surfaces across all viewing angles, which are separately positioned at right, top, right-top, and right-bottom relative to the object. Besides, we select six diverse HDRI environment maps of both natural and artificial lighting conditions in indoor and outdoor scenarios, as shown in Figure S2. For each lighting configuration, we generate a 40-frame

---

[3]Poly Haven website: `https://polyhaven.com/hdris`

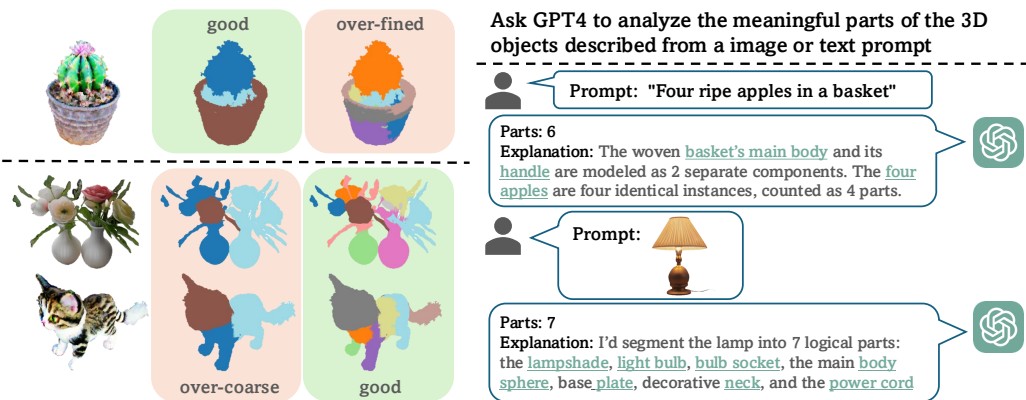

Figure S1: **Visualization about 3D part segmentation. Left**: Due to varying complexity across prompts, assigning a fixed number of parts to all objects is suboptimal. **Right**: We propose to estimate prompt-specific part counts with GPT4V to better reflect meaningful structural granularity.

video sequence with the object rotating 360 degrees. This rigorous relighting protocol establishes a robust and comprehensive representation of material characteristics that closely approximates real-world viewing conditions. Qualitative results are visualized in Figure S3.

## A.3 Annotation pipeline

In this section, we provide a detailed introduction to our automatic annotation system ($M^2$AP) and present related experiments concerning its design and validation.

**Selected agents.** To automatically provide practical and human-aligned scores for each 3D asset, we employ advanced multi-modal large language models (MLLMs) as our labeling agents. For objective and fair evaluations, preliminary experiments led to the selection of two types of models as scoring experts: thinking models and reasoning models. Thinking models excel at deep reflection and analysis, while reasoning models leverage extensive knowledge bases to deliver more efficient and stable results. The selected models include GPT 4.1 ( https://openai.com/), GPT o3/o4 mini ( https://openai.com/), Gemini 2.5 Pro ( https://gemini.google.com/), Claude 3.7 ( https://www.anthropic.com/), and Grok-3 ( https://x.ai/). Gemini 2.5 Pro processes rotating videos of 3D objects as input, while other agents process multi-view images.

**Prompt design.** To mitigate potential MLLM hallucination and ensure accurate, consistent 3D asset evaluations, we engineered an elaborate prompt. The prompt guides the MLLMs through a systematic process, defines clear assessment criteria, incorporates physical realism checks, includes a reflection phase, and uses comparative examples to align automated scoring with human perceptual judgments.

*Stepped instruction.* The MLLM is first assigned the role of an expert evaluator, tasked with providing a detailed initial description of the 3D asset. It then systematically analyzes and follow a step-by-step instruction the asset from multiple views (Geometry, Normal Map, RGB) and perspectives, aiming for a comprehensive understanding. The findings are structured into a predefined JSON output, ensuring a methodical evaluation flow.

*Criterion definition.* For consistent and objective scoring, the prompt specifies a set of key dimensions: Geometry Plausibility (GP), Geometry Details (GD), Texture Quality (TQ), Geometry-Texture Coherency (GTC), Prompt-Asset Alignment (PA), Details and Complexity (DC), Colorfulness and Saturation (CS), Consistency and Artifacts (CA), and Material Plausibility (MP). As illustrated in Figure S4 and Figure S5, each dimension is equipped with a multi-level scoring rubric, supported by clearly defined qualitative descriptors and a structured evaluation protocol. This enables MLLMs to systematically assess the 3D content by comparing it with standardized quality levels.

*Physical alignment.* To ensure generated assets are realistic, the prompt emphasizes assessing physical plausibility. It involves checking the correct positioning of object parts and evaluating if material properties (e.g., smoothness of wood or metal) align with real-world expectations. Structural anomalies or incorrect physical characteristics are penalized.

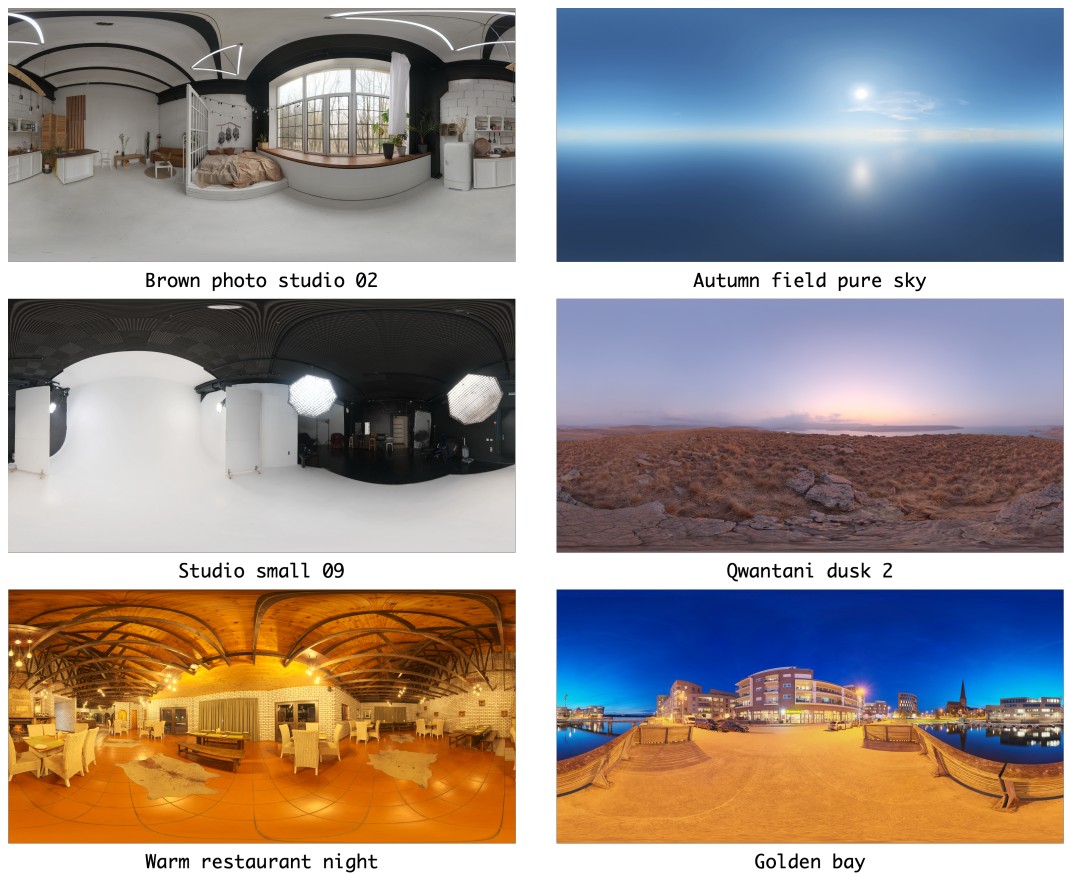

Brown photo studio 02

Autumn field pure sky

Studio small 09

Qwantani dusk 2

Warm restaurant night

Golden bay

Figure S2: **Visualization about HDRI maps.** We select six HDRI maps from Poly Haven[3], including natural, artificial, indoor, outdoor, daylight and nightlight conditions.

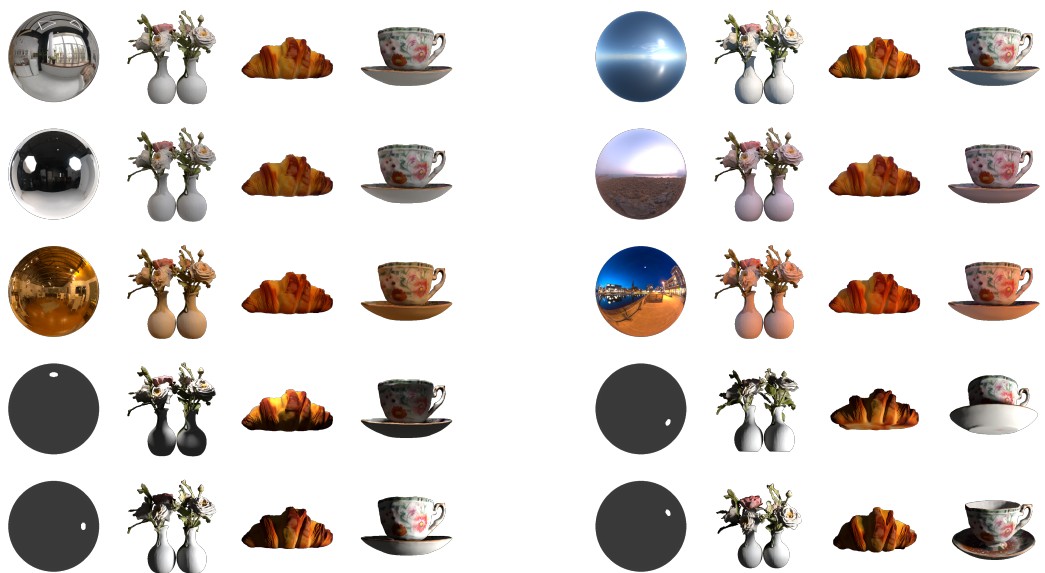

Figure S3: **Visualization about relighting.** We present the first frame of each object under varying illumination conditions. The leftmost metallic sphere serves as a reference, reflecting the HDRI environment or point light source position. Additionally, for the right-top and right-bottom light configurations, we adjust the camera elevation to ensure full object coverage.

Table R1: **Ablation study about the annotation pipeline.** We conduct a systematic comparison between our proposed annotation pipeline and baseline approaches with single LLM agents, employing the L1 loss between model outputs and human judgments as an evaluation metric. Through component-wise ablation studies, we further analyze each key element in our pipeline design.

| Method | Single Agent | | | | | M²AP | | |
| --- | --- | --- | --- | --- | --- | --- | --- | --- |
| | GPT 4.1 | Claude 3.7 | Gemini 2.5 | Grok 3 | o3/o4 mini | w/o Physical | w/o Reflection | **Full** |
| **L1 Loss ↓** | 0.838 | 1.100 | 1.020 | 0.920 | 0.702 | 0.568 | 0.476 | **0.257** |

Table R2: **API cost for annotation procedures**.

| Annotation type | Object-level | Part-level | Material-subject | Total |
| --- | --- | --- | --- | --- |
| **Cost (USD)** | 2.5k | 1.0k | 0.6k | 4.1k |

*Reflection.* A self-correction phase is integrated to enhance reliability. The MLLM, acting as an autonomous auditor, re-evaluates its initial textual analysis and scores against the predefined criteria, ignoring its prior numerical assignments. If discrepancies are found, it provides revised scores, aiming to improve the accuracy and consistency of the final assessment.

*Typical examples.* To align automated scores with human perception, the prompt includes examples comparing human annotations with typical MLLM evaluations. It guides the agents to understand human evaluators' focus, calibrate its scoring to reflect human priorities and sensitivities to flaws or strengths, and reduce biases, ultimately capturing nuances important to human judgment.

The structured prompt design aims to minimize ambiguity and ensure MLLMs adhere to a rigorous, consistent, and human-aligned evaluation.

**Ablation study.** Our experiments validate the effectiveness of the proposed $M^2$AP annotation pipeline through systematic comparisons with individual MLLM agents and component-wise ablations, using L1 distance to human annotations as the evaluation metric. As shown in Table R1, the complete $M^2$AP framework achieves superior performance (L1=0.257), significantly outperforming individual state-of-the-art models. Furthermore, ablations demonstrate the critical role of each component: omission of either the Physical Alignment check (L1=0.568) or Reflection mechanism (L1=0.476) substantially degrades performance, confirming their importance for annotation accuracy.

**Annotation Cost.** Without parallelization, annotating a single object typically takes around 20 to 60 seconds, depending mainly on the network latency. In terms of cost, completing one M²AP annotation-which involves calls to multiple VLM APIs—for a single object incurs approximately 0.15 USD. Statistics of the cost for each setting are shown in Table R2.

**Human Validation** To assess the validity and reliability of the proposed automated annotation pipeline and scoring models, we collected a set of human annotations for empirical validation.

- **Open-source annotations for object-level evaluation.** Under the same standardized criteria for object-level evaluation, we directly adopt human annotations from 3DGen-Bench [73]. According to its thesis, the 3DGen-Bench team employed 47 professional annotators via a crowdsourcing platform. To ensure annotation quality, they provided detailed guidelines, conducted regular monitoring, and proposed a "Rank-and-Rate" protocol: annotators first rank assets generated from the same prompt, then assign dimension-wise scores. Each asset is independently evaluated by two annotators, and multiple validation strategies are applied to clean up the data. Finally, we sampled 87 text prompts and 86 image prompts, yielding 1,210 annotated assets.

- **In-house user study for part- and material- level evaluation.** Since existing benchmarks focus only on object-level evaluation, we conducted a dedicated user study for part- and material-level assessment. We recruited a total of 8 expert human annotators (5 females and 3 males), all of whom are Ph.D. students with prior experience in 3D modeling or evaluation, ensuring a solid understanding of the assessment criteria. To promote scoring consistency, we designed a detailed annotation protocol (expanded from prompt templates illustrated in Figures S4 and S5), which includes explicit definitions and example visualizations for each evaluation dimension. All annotators also underwent a calibration phase before annotation.

Table R3: **Ablation study about the video-based scoring model.** We evaluate the L1 loss of scores and the accuracy of pairwise rating in the DC dimension under several settings. Our final configuration selects CLIP as the prompt encoder, sets the dropout ratio to 0.5, and combines SmoothL1 loss and ranking loss as the objective function.

| | Prompt Encoder DINOv2 | Dropout Ratio 0.1 | Objective Function MSE | MAE | **ours** |
|---|---|---|---|---|---|
| **L1 Loss** ↓ | 0.550 | 0.426 | 0.332 | 0.397 | **0.312** |
| **Pairwise Accuracy** ↑ | 0.621 | 0.708 | 0.758 | 0.757 | **0.798** |

Table R4: **Ablation study on video frames.** We calculate L1 loss for each dimension under different frame number settings. The average inference time per object is listed below.

| Frames | DC | CS | CA | MP | Inference Time ($s$ / $it$) |
|---|---|---|---|---|---|
| 4 | 0.338 | 0.330 | 0.341 | 0.343 | 0.211 |
| 8 | 0.335 | 0.311 | 0.337 | 0.323 | 0.250 |
| 32 | 0.311 | 0.296 | 0.294 | 0.291 | 0.497 |
| 16 (Ours) | 0.312 | 0.291 | 0.288 | 0.287 | 0.320 |

To reduce individual bias, each sample was independently rated by at least 3 annotators, and the final score is obtained by averaging individual ratings. In practice, the annotations are collected via structured questionnaires. For material-level annotations, we follow the "Rank-and-Rate" strategy, which encourages comparative judgment, helping annotators develop a more consistent internal scale and reducing scoring drift across samples. Finally, we select 25 test prompts and sample 3-4 assets per prompt, resulting in 86 annotated assets. For part-level annotation, we sample 40 test assets and select 3-4 parts from each, yielding 159 annotated parts.

# B   More details about video-based scoring model

## B.1   Ablation experiments

**Prompt encoder.** Due to the different performance of CLIP [42] and DINOv2 [39] image encoders in 3D awareness, we investigate the effect of image encoders in training stage-1. As shown in Table R3, there exists a clear decrease in scoring accuracy when DINOv2 is employed as the image prompt encoder. One potential explanation is that the CLIP text encoder is selected as the text prompt encoder, which provides better alignment for the CLIP image encoder in the latent space, leading to more effective training outcomes in stage-1.

**Dropout ratio.** Our prediction head incorporates Dropout layers followed by two Conv3D layers to mitigate overfitting, as depicted in Figure 3. Through the ablation study in Table R3, we demonstrate that increasing the dropout ratio not only accelerates training convergence but also enhances inference accuracy. This suggests that aggressive dropout is particularly effective when processing large-scale video features extracted by the encoder, likely due to its capacity to robustly regularize high-dimensional spatio-temporal representations.

**Objective function.** As described in 4.1, our final loss function is composed of Smooth L1 Loss and Rank Loss. To examine the effectiveness of our loss function, we conduct ablation experiments in which prediction heads are trained using different losses in DC dimension. Table R3 reveals that MAE falls short in penalty for large errors compared to MSE and ours. For pairwise rating accuracy, our ranking loss obviously contributes to the relative comparison capability of the prediction head, which demonstrates the effectiveness of our loss design.

**Frame count.** We carry out experiments on different frame numbers of input videos. As shown in Table R4, with frames increasing from 4 to 16, the scoring accuracy also shows a consistent upward trend. However, there is no significant difference between 16 frames and 32 frames in the aspect of accuracy, which indicates abundant information is included in 16 frames. Considering the trade-off between accuracy and inference efficiency, we adopt 16 frames as the final setting.

Table R5: **Pairwise rating alignment with $T^3$Bnech at the object level.** Image Reward, $T^3$Bnech, and GPTEval3D could only calculate the pairwise scores among text-to-3D objects.

| Metrics | Text-to-3D | | | | | Image-to-3D | | | | |
|---|---|---|---|---|---|---|---|---|---|---|
| | GP | GD | TQ | GTC | PA | GP | GD | TQ | GTC | PA |
| CLIP Score [42] | 0.556 | 0.580 | 0.606 | 0.556 | 0.604 | 0.589 | 0.588 | 0.605 | 0.636 | 0.623 |
| ViCLIP Score [55] | 0.557 | 0.591 | 0.625 | 0.577 | 0.617 | 0.589 | 0.570 | 0.611 | 0.640 | 0.623 |
| Aesthetic Score [44] | 0.657 | 0.634 | 0.607 | 0.629 | 0.623 | 0.570 | 0.613 | 0.622 | 0.675 | 0.630 |
| Image Reward [65] | 0.568 | 0.598 | 0.607 | 0.513 | 0.610 | - | - | - | - | - |
| T3Bench [13] | 0.661 | 0.647 | 0.628 | 0.673 | 0.631 | - | - | - | - | - |
| GPTEval3D [61] | 0.690 | 0.689 | 0.677 | 0.667 | 0.649 | - | - | - | - | - |
| **Ours** | **0.774** | **0.725** | **0.755** | **0.749** | **0.726** | **0.718** | **0.703** | **0.753** | **0.732** | **0.710** |

Table R6: **Full leaderboard at the object level.** We accumulate the scores of five dimensions as the overall score and sort methods in the sequence of overall performance.

| Method | Method Type | GP | GD | TQ | GTC | PA | Overall |
|---|---|---|---|---|---|---|---|
| Hunyuan3D 2.5 [25] | Image-to-3D | 6.46 | 2.86 | 2.79 | 0.981 | 3.47 | 16.561 |
| Hunyuan3D 2.0 [74] | Image-to-3D | 6.2919 | 2.7215 | 2.7644 | 0.9876 | 3.4334 | 16.1988 |
| Hunyuan3D 2.5 [25] | Text-to-3D | 6.42 | 2.7 | 2.45 | 0.947 | 3.18 | 15.697 |
| Trellis [63] | Image-to-3D | 5.8626 | 2.392 | 2.4693 | 0.9702 | 3.5048 | 15.1989 |
| SPARD3D [19] | Image-to-3D | 5.7791 | 2.3031 | 2.4749 | 0.9601 | 3.4842 | 15.0014 |
| TripoSR [52] | Image-to-3D | 5.2216 | 2.4225 | 2.3758 | 0.9562 | 3.3643 | 14.3404 |
| InstantMesh [64] | Image-to-3D | 5.4242 | 2.2252 | 2.3063 | 0.9587 | 3.363 | 14.2775 |
| CRM [58] | Image-to-3D | 4.745 | 2.2991 | 2.3777 | 0.9164 | 3.219 | 13.5572 |
| MVdream [45] | Text-to-3D | 4.4064 | 2.742 | 2.8116 | 0.951 | 2.5879 | 13.4989 |
| Unique3D [60] | Image-to-3D | 4.9288 | 2.3233 | 1.9627 | 0.776 | 3.1989 | 13.1897 |
| OpenLRM [14] | Image-to-3D | 3.7754 | 2.2614 | 2.0922 | 0.902 | 2.2298 | 11.2608 |
| Wonder3D [34] | Image-to-3D | 3.7879 | 2.0092 | 1.9658 | 0.9255 | 2.0874 | 10.7758 |
| Stable-Zero123 [32] | Image-to-3D | 3.6052 | 1.6548 | 2.0293 | 0.8902 | 2.2578 | 10.4374 |
| Magic123 [41] | Image-to-3D | 3.4617 | 1.74 | 2.0094 | 0.898 | 2.2171 | 10.3262 |
| GRM-Image [66] | Image-to-3D | 3.2932 | 1.857 | 1.8885 | 0.849 | 2.0735 | 9.9612 |
| LGM [50] | Image-to-3D | 3.2148 | 1.6733 | 1.8891 | 0.8118 | 2.0304 | 9.6193 |
| Lucid-Dreamer [28] | Text-to-3D | 2.9346 | 1.5891 | 2.1069 | 0.8333 | 2.0297 | 9.4936 |
| GRM-Text [66] | Text-to-3D | 3.0096 | 1.6627 | 1.7389 | 0.898 | 1.793 | 9.1023 |
| Latent-NeRF [36] | Text-to-3D | 2.7265 | 1.7067 | 1.7065 | 0.8412 | 1.7688 | 8.7497 |
| Magic3D [29] | Text-to-3D | 2.9015 | 1.6239 | 1.5395 | 0.9431 | 1.5618 | 8.5698 |
| SyncDreamer [33] | Image-to-3D | 2.9423 | 1.5323 | 1.2134 | 0.8529 | 1.2776 | 7.8185 |
| Dreamfusion [40] | Text-to-3D | 2.669 | 1.2525 | 1.183 | 0.9137 | 1.3446 | 7.3627 |
| Triplane-Gaussian [77] | Image-to-3D | 2.2948 | 1.1859 | 1.2028 | 0.6647 | 1.2908 | 6.6389 |

## B.2 Comparison with T3Bench

To further validate the effectiveness of our scoring framework, we conduct a supplementary experiment comparing pairwise rating alignment with $T^3$Bench [13], a benchmark designed for evaluating text-to-3D generation. Specifically, we follow the standard pairwise protocol used in Section 5.1. As reported in Table R5, our method achieves a significantly higher alignment with human judgments compared to $T^3$Bench [13], providing more reliable discrimination ability in pairwise comparisons.

## B.3 Leaderboard

**Object level.** Table R6 presents the comprehensive leaderboard for object-level evaluation across 22 methods, including image-condition and text-condition methods. Hunyuan3D 2.5 achieves the highest overall performance (16.561), outperforming other approaches across most dimensions, particularly in Geometry Plausibility (6.46). Image-to-3D methods generally dominate the upper rankings, with Hunyuan3D, Trellis, and SPARD3D forming the top three.

**Material subject.** Based on the object-level evaluation, we select 23 methods that demonstrate acceptable performance in texture and geometry. The overall leaderboard is shown in Table R7, suggesting great potentials for text-to-shape methods and space for improvement in aspects of texture detail and visual harmony.

Table R7: **Full leaderboard at the material level.** We accumulate the scores of four dimensions as the overall score and sort methods in the sequence of overall performance.

| Method | Method Type | DC | CS | CA | MP | Overall |
|---|---|---|---|---|---|---|
| Hunyuan3D 2.0 [74] | Image-to-3D | 2.5332 | 3.0001 | 3.0832 | 2.9344 | 11.5509 |
| Trellis [63] | Image-to-3D | 2.4812 | 3.0014 | 3.0138 | 2.9036 | 11.4000 |
| SPAR3D [19] | Image-to-3D | 2.4742 | 3.0561 | 2.6510 | 2.6756 | 10.8568 |
| TripoSR [52] | Image-to-3D | 2.3262 | 2.7841 | 2.7001 | 2.6832 | 10.4936 |
| InstantMesh [64] | Image-to-3D | 2.1675 | 2.7699 | 2.6279 | 2.5735 | 10.1388 |
| Wonder3D [34] | Image-to-3D | 1.9707 | 2.5512 | 2.5148 | 2.3972 | 9.4339 |
| CRM [58] | Image-to-3D | 2.0305 | 2.6174 | 2.3580 | 2.3836 | 9.3896 |
| Mvdream [45] | Text-to-3D | 2.0029 | 2.5179 | 2.4175 | 2.2525 | 9.1907 |
| LN3Diff [26] | Image-to-3D | 1.9392 | 2.5072 | 2.3494 | 2.3432 | 9.1390 |
| GRM-text [66] | Text-to-3D | 1.7664 | 2.4341 | 2.5874 | 2.3500 | 9.1378 |
| Magic3d [29] | Text-to-3D | 1.7630 | 2.2465 | 2.6972 | 2.0950 | 8.8018 |
| OpenLRM [14] | Image-to-3D | 1.9777 | 2.4608 | 2.0886 | 2.2672 | 8.7943 |
| LucidDreamer [28] | Text-to-3D | 1.8763 | 2.3311 | 2.3131 | 2.1951 | 8.7156 |
| Dreamfusion [40] | Text-to-3D | 1.5700 | 2.2032 | 2.7198 | 2.1211 | 8.6141 |
| LGM [50] | Image-to-3D | 1.7470 | 2.3838 | 2.2318 | 2.1953 | 8.5579 |
| GRM-image [66] | Image-to-3D | 1.6203 | 2.3085 | 2.4136 | 2.1629 | 8.5052 |
| Stable Zero123 | Image-to-3D | 1.8004 | 2.3785 | 2.1614 | 2.0483 | 8.3887 |
| Zero-1-to-3 [32] | Image-to-3D | 1.7317 | 2.3282 | 2.2199 | 2.0657 | 8.3454 |
| 3DTopia-XL [5] | Image-to-3D | 1.6359 | 2.2525 | 1.9778 | 1.9235 | 7.7898 |
| SyncDreamer [33] | Image-to-3D | 1.2629 | 2.0990 | 2.4309 | 1.8567 | 7.6494 |
| Latent-Nerf [36] | Text-to-3D | 1.4644 | 1.9590 | 2.0826 | 1.7380 | 7.2440 |
| Triplane [77] | Image-to-3D | 1.1629 | 1.9399 | 2.0471 | 1.7167 | 6.8665 |
| SJC [54] | Text-to-3D | 0.9066 | 1.2189 | 2.0290 | 1.1470 | 5.3016 |

Table R8: **Ablation experiments of the 3D-based scoring model.** We conduct ablation experiments for proposed attention modules and predict heads with the criterion of normalized L1 and L2 loss on the test set. The setting of our model is composed of two attention modules for global-part interaction and inner-part interaction, and the predict head is a Linear layer.

| | **ours** | w/o global_attn | w/o part_attn | w/o attns | 2-layer mlp | 3-layer mlp |
|---|---|---|---|---|---|---|
| **L1 Loss** ↓ | **0.0850** | 0.0866 | 0.0935 | 0.0913 | 0.0903 | 0.0877 |
| **L2 Loss** ↓ | **0.0115** | 0.0127 | 0.0142 | 0.0135 | 0.0131 | 0.0127 |

# C  More details about 3D-based scoring model

## C.1  Ablation experiments

We conduct ablation experiments for our proposed 3D-based scoring model on 493 test parts, including the two attention modules and the depth of MLP, as described in Section 4.2.

**Attention modules.** As shown in Table R8, experimental results clearly demonstrate the complementary roles of the cross-attention and self-attention modules in part-level quality prediction. The cross-attention mechanism effectively integrates global contextual cues into each part representation, while the self-attention module plays a crucial role in capturing intra-part dependencies and enforcing local coherence. Removing the self-attention module leads to a significant drop in performance, indicating the essentiality of modeling inner-part interactions.

**Predict head.** To investigate the impact of MLP depth in the prediction head, we conduct an ablation study varying the number of layers from 1 to 3. As summarized in Table R8, the predictor achieves the best performance with a single-layer MLP. Interestingly, using a 3-layer MLP yields slightly worse results, while the 2-layer variant performs the worst among all settings. We hypothesize that a deeper MLP may introduce unnecessary complexity and overfitting risks. These findings suggest that a simple 1-layer MLP strikes a better balance between capacity and generalization for this task.

# Object-Level Criteria

**1) Geometry Plausibility**: Assess using Ge Normal Map & RGB View;

**Please strictly adhere to this scoring criteria: Award high scores (6, 7, 8) to truly excellent works that demonstrate exceptional quality, detail and craftsmanship. Give low scores (0, 1, 2) to poor quality 3D models. Avoid always giving "safe" average scores (3, 4, 5) unless the work is decently good and objectively average in quality. Be rigorous in your evaluation - reward excellence without hesitation and penalize substandard work accordingly. You should tolerate some minor imperfections. Maintain scoring consistency across all assessments, following the below scoring standards. Consider the position and the physical properties of the part in the object. Wrong position should be penalized. Smoothness of some physical properties (like wooden or ironwork materials) should be acceptable. Remember if the object has the normal and recognizable structure, consider give the score higher than 5.**

- 0: (Low Scores) Complete collapse/blank (training failure).
- 1-2: (Low Scores) Unrecognizable or nonsensical shapes (e.g., fragmented geometry, severe abstraction).
- -- 1: (Low Score) Barely recognizable fragments: random directions (Normal) (e.g., vague outlines but no coherent structure).
- -- 2: (Low Score) Abstract shapes unrelated to the prompt: chaotic patterns (Normal) (e.g., "cat" rendered as chaotic geometry).
- 3-5: (Middle-Range Scores) Recognizable in BOTH views but flawed:
- -- 3: (Middle Score) Clear object identity but severe issues: Structural anomalies visible in BOTH views (e.g., clear multi-head issue, structural anomalies, wrong position).
- -- 4: (Middle Score) Structurally normal but overly simplistic: basic Normals (e.g., basic shape with no details, minimal noise).
- -- 5: (Middle Score) Structurally normal with some clear noise/details: logical Normals (minor surface noise allowed) (e.g., identifiable "chair" with surface bumps but no defects).
- 6-8: (High-Range Scores) Structurally sound + detail-rich:
- -- 6: (High Score) Clean structure + minor and not clear defects: accurate Normal details (e.g., small surface dents).
- -- 7: (High Score) High-quality details + minimal and barely visible defects/noise: nuanced Normals (e.g., intricate carvings).
- -- 8: (High Score) Very high-quality: sophisticated Normal details and ignorable defects (for exceptional cases).
- 9: Unused (unattainable by current models).

**2) Geometry Details**: Use Geometry & Normal Map & RGB View (ignore plausibility, focus on detail density)

**Please strictly adhere to this scoring criteria: Award high scores (3, 4) to truly excellent works that demonstrate exceptional quality, detail and craftsmanship. Give low scores (0, 1) to poor quality 3D models. Avoid always giving "safe" average scores (2) unless the work is decently good and objectively average in quality. Be rigorous in your evaluation - reward excellence without hesitation and penalize substandard work accordingly. You should tolerate some minor imperfections. Maintain scoring consistency across all assessments, following the below scoring standards.**

- 0: (Low Score) Blank/training failure (e.g., entirely black or no structure).
- 1: (Low Score) Smooth surfaces, no meaningful details (only basic shapes for recognition, e.g., cat ears/paws).
- 2: (Middle Score) Minimal details with possible noise (e.g., simple facial features like eyes/mouth, but blurry or noisy).
- 3: (High Score) Moderate details (clear features like whiskers, texture folds; minimal noise).
- 4: (High Score) Highly detailed (complex structures like fur, ornaments; near-realistic density, negligible noise).

**3) Texture Quality**: Focus on RGB View

**Please strictly adhere to this scoring criteria: Award high scores (3, 4) to truly excellent works that demonstrate exceptional quality, detail and craftsmanship. Give low scores (0, 1) to poor quality 3D models. Avoid always giving "safe" average scores (2) unless the work is decently good and objectively average in quality. Be rigorous in your evaluation - reward excellence without hesitation and penalize substandard work accordingly. You should tolerate some minor imperfections. Maintain scoring consistency across all assessments, following the below scoring standards.**

- 0: No texture/extreme blur (unrecognizable)
- 1: Low aesthetic (blurry but recognizable base features)
- 2: Decent aesthetic (partial clarity, inconsistent across views)
- 3: High aesthetic (consistent style across views)
- 4: Photorealistic (flawless, view-consistent)

**4) Prompt-Asset Alignment**: Compare with reference

**Please strictly adhere to this scoring criteria: Award high scores (3, 4) to truly excellent works that demonstrate exceptional quality, detail and craftsmanship. Give low scores (0, 1) to poor quality 3D models. Avoid always giving "safe" average scores (2) unless the work is decently good and objectively average in quality. Be rigorous in your evaluation - reward excellence without hesitation and penalize substandard work accordingly. You should tolerate some minor imperfections. Maintain scoring consistency across all assessments, following the below scoring standards.**

- 0: Unrelated/unrecognizable
- 1: Partial match (correct category only)
- 2: Majority match (key attributes correct; quantity/position errors)
- 3: Near-complete match (minor detail deviations)
- 4: Perfect alignment (all textual/image elements precisely replicated)

**5) Geometry-Texture Coherency**: Compare all views
- 0: Inconsistent (texture masks geometry flaws/conflicts)
- 1: Coherent (texture aligns naturally with geometry)

Figure S4: **Scoring criterion of each dimension at the object level.**

# Part-Level Criteria

**1) Geometry Plausibility**: Assess using Normal Map & RGB View;

**Please strictly adhere to this scoring criteria: Award high scores (6, 7, 8) to truly excellent works that demonstrate exceptional quality, detail and craftsmanship. Give low scores (0, 1, 2) to poor quality 3D models. Avoid always giving "safe" average scores (3, 4, 5) unless the work is decently good and objectively average in quality. Be rigorous in your evaluation - reward excellence without hesitation and penalize substandard work accordingly. You should tolerate some minor imperfections. Maintain scoring consistency across all assessments, following the below scoring standards. Consider the position and the physical properties of the part in the object. Wrong position should be penalized. Smoothness of some physical properties (like wooden parts) should be acceptable.**

- 0: (Low Scores) Complete collapse/blank (training failure).
- 1-2: (Low Scores) Unrecognizable or nonsensical shapes (e.g., fragmented geometry, severe abstraction).
-- 1: (Low Score) Barely recognizable fragments: random directions (Normal) (e.g., vague outlines but no coherent structure).
-- 2: (Low Score) Abstract shapes unrelated to the prompt: chaotic patterns (Normal) (e.g., "cat" rendered as chaotic geometry).
- 3-5: (Middle-Range Scores) Recognizable in BOTH views but flawed:
-- 3: (Middle Score) Clear object identity but severe issues: Structural anomalies visible in BOTH views (e.g., clear multi-head issue, structural anomalies, wrong position).
-- 4: (Middle Score) Structurally normal but overly simplistic: basic Normals (e.g., basic shape with no details, minimal noise).
-- 5: (Middle Score) Structurally normal with some clear noise/details: logical Normals (minor surface noise allowed) (e.g., identifiable "chair" with surface bumps but no defects).
- 6-8: (High-Range Scores) Structurally sound + detail-rich:
-- 6: (High Score) Clean structure + minor and not clear defects: accurate Normal details (e.g., small surface dents).
-- 7: (High Score) High-quality details + minimal and barely visible defects/noise: nuanced Normals (e.g., intricate carvings).
-- 8: (High Score) Very high-quality: sophisticated Normal details and ignorable defects (for exceptional cases).
- 9: Unused (unattainable by current models).

**2) Geometry Details**: Use Normal Map & RGB View (ignore plausibility, focus on detail density)

**Please strictly adhere to this scoring criteria: Award high scores (3, 4) to truly excellent works that demonstrate exceptional quality, detail and craftsmanship. Give low scores (0, 1) to poor quality 3D models. Avoid always giving "safe" average scores (2) unless the work is decently good and objectively average in quality. Be rigorous in your evaluation - reward excellence without hesitation and penalize substandard work accordingly. You should tolerate some minor imperfections. Maintain scoring consistency across all assessments, following the below scoring standards.**

- 0: (Low Score) Blank/training failure (e.g., entirely black or no structure).
- 1: (Low Score) Smooth surfaces, no meaningful details (only basic shapes for recognition, e.g., cat ears/paws).
- 2: (Middle Score) Minimal details with possible noise (e.g., simple facial features like eyes/mouth, but blurry or noisy).
- 3: (High Score) Moderate details (clear features like whiskers, texture folds; minimal noise).
- 4: (High Score) Highly detailed (complex structures like fur, ornaments; near-realistic density, negligible noise).

# Material-Level Criteria

**1) Detail & Complexity**: Focus Geometry View and RGB View (*ignore geometry defects, focus on details*)

This criterion assesses the richness of the texture in terms of its visual details and complexity while maintaining a harmonious balance to prevent overly fancy patterns that may compromise aesthetics. Judgements should primarily focus on *Level of Detail* (Does the texture offer enough fine details that could enhance realism, or is it too simplistic?) and *Balance of Simplicity and Complexity* (Is there a good balance between intricate details and simple areas, avoiding visual overload?)

- 0: No clear features of textures, no observable detail resolution
- 1: Very simplistic or overly detailed, with subtle patterns or excessive ornamentation
- 2: Poor balance between simplicity and intricacy, like inbalance between different surfaces, imperfect or in improper positions.
- 3: Moderate details with only little defects that slightly affect the overall impressions
- 4: Suitable details with an optimal visual balance, without any visible defects

**2) Colorfulness & Saturation**: Focus Geometry View and RGB View (*ignore geometry defects, focus on colors*)

This evaluates the overall color distribution across the texture and its visual clarity at a glance. Judgements should primarily focus on *Color Diversity* (Are there proper color variations to distinguish features or does the texture look too monotone?), *Saturation* (Are the colors appropriately saturated, neither too muted nor too vibrant?) and *Color Suitability* (Does the hue logically match the reality well and improve its visual effects?)

- 0: No clear features of textures, no discernible color features
- 1: Extremely saturated or colorful, or with an irregular color distribution, which confounds the perceptual clarity
- 2: Incorrect saturation or lack of color diversity, like a monochromatic appearance that does not suit the object
- 3: Suitable saturation with only little color defects, like only little colors that mismatch the reality
- 4: Excellent colorfulness and easy to interpret, without noticeable color faults

**3) Consistency & Artifact**: Focus all Lighting Views

This criterion evaluates whether the texture remains uniform and cohesive under different lighting conditions (e.g., indoor / outdoor). Judgements should primarily focus on whether there are *Noticeable Seams or Borders* (specifically whether abrupt changes in shading that break the texture's flow under varying lighting conditions) and *Shading Artifacts* (Are there unnatural lighting streaks or shadows on the surface when illuminated? Are there unreasonable specular highlights appearing when the surface is in dark?)

- 0: No clear features of textures under varying illumination conditions
- 1: Large-scale discontinuity such as large area of luminance differences / incorrect highlights / black shadows
- 2: Clear inconsistencies or artifacts in multi-views, such as abrupt brightness changes
- 3: Only little inconsistencies or shading artifacts which are very imperceptible
- 4: Perfectly consistent and seamless without shading artifacts

**4) Material Plausibility**: Focus indoor and outdoor Lighting Views

This evaluates whether the texture exhibits plausible diffuse and specular lighting effects that align with the real-world material behaviors as specified in the prompt. Judgements should primarily focus on *Metalness* (Does the texture display physically suitable specular lighting effects under natural lighting condition?) and *Roughness* (Does it correctly represent diffuse lighting effects?)

- 0: No clear features of textures, no discernible structural features
- 1: Poor reflection behaviors or extreme exposure due to both incorrect metalness and roughness
- 2: Incorrect metallic or roughness settings, or unsuitable exposure that reduces visual effects
- 3: Only little difference from material specification in prompt
- 4: Excellent light reflection with plausible exposure and aligns well with reality

Figure S5: **Scoring criterion of each dimension at the part level and material level.**

