# OpenReview forum: "Hi3DEval: Advancing 3D Generation Evaluation with Hierarchical Validity"
_NeurIPS.cc/2025/Datasets_and_Benchmarks_Track — NeurIPS 2025 Datasets and Benchmarks Track poster_

### Official Review · Reviewer_7bHR · 2025-06-29

**Rating:** 5
**Confidence:** 4

**Summary:**

This paper proposes Hi3DEval, a hierarchical framework for evaluating 3D generative models. It addresses limitations in current 3D evaluation methods by introducing a three-level scoring mechanism—object-level, part-level, and material-level—using both human-aligned annotations and automated hybrid scoring models. The authors also contribute Hi3DBench, a large-scale dataset with multi-agent, multi-modal annotations, and introduce novel video- and geometry-aware scoring pipelines tailored for assessing various facets of 3D quality.

**Additional Feedback:**

1.Will the inference weights of the scoring system open source, or provide an evaluation platform to facilitate the community in assessing the quality of 3D generation?
2.Do we need to perform 3D-based scoring for the complete mesh in order to better assess the overall geometric quality?

**Dataset Code Accessibility:**

Yes

**Dataset Code Comments:**

The dataset and code are open access and could be downloaded from Huggingface and Github.

**Ethical Considerations:**

No, there are no or only very minor ethics concerns

**Final Justification:**

The rebuttal resolves my concerns, and I appreciate the detailed response. I will maintain the positive score.

**Limitations Weaknesses:**

1.The assumption of reliable mesh segmentation and semantically consistent parts across diverse object categories is fragile and can fail on deformable or abstract objects (e.g., animals, organic forms).
2.Among all these evaluation metrics, which ones contribute more significantly, which ones are suitable for what kinds of applications? There is a lack of analysis in this area.
3.The paper omits details on rendering details, such as rendering tools of Blender or other tools, and resolutons of the images, will different qualities of the images affect the scoring system?
4.The composition of the 15,000 assets is not detailed, raising concerns about diversity, representativeness, and potential biases in the dataset.

**Strengths Contributions:**

1.Hierarchical Evaluation Framework: The framework’s multi-granularity scoring (object, part, material) enables detailed diagnosis of 3D generation failures, going well beyond existing benchmarks that offer only coarse object-level metrics.
2.Hybrid Scoring Models: The combination of video-based encoders for spatio-temporal features and 3D mesh-based models (via PartField) is a strong technical contribution, particularly in part-level flaw localization.
3.Automated Annotation Pipeline: M²AP utilizes LLMs and multimodal agents for consistent, scalable annotations, minimizing subjectivity and manual labor, with human-agent alignment validated through correlation studies.
4.Scalable Dataset: Hi3DBench provides 15,000+ diverse 3D assets with annotations at object, part, and material levels, enabling robust evaluation across multiple dimensions.
5.Empirical Superiority: The method consistently outperforms strong baselines like GPTEval3D and CLIPScore across all levels of evaluation in pairwise human alignment.

---

> ### Author Rebuttal · Authors · 2025-07-30
>
> Thank you for taking the time to review our work and for providing clear and helpful feedback. We appreciate your thoughtful comments, which have helped us refine both the presentation and analysis. Below, we respond to each of your points in detail
>
> ### Q1: Discussion on 3D part segmentation
> We agree that reliable part segmentation can be challenging for deformable or abstract objects. However, **PartField represents the state-of-the-art** among current category-agnostic part segmentation methods in terms of both quality and scalability (as discussed in Section A.2). Moreover, since part segmentation serves as a preprocessing step in our pipeline, our framework is compatible with future improvements—and **a more reliable method can be easily substituted when available.**
>
> In the context of our evaluation task, **the primary role of segmentation** is to help localize flawed regions in the generated 3D assets. As such, **some degree of segmentation noise or imperfection is tolerable**, as long as the overall part structure remains coherent enough to support perceptual analysis.
>
> Furthermore, for assets that are **too abstract**, **severely deformed**, or **clearly fail in generation**, we hold the opinion that conducting part-level evaluation on them is not meaningful.  In such cases, it is more appropriate to **focus on object-level improvements rather than analyzing localized regions**. Therefore, we suggest excluding such instances from part-level evaluation, as doing so helps maintain evaluation reliability and interpretability.
>
> ### Q2: Analysis for evaluation dimensions
> Thank you for raising this important point. Indeed, different evaluation metrics serve distinct purposes and carry varying weights depending on the target application. Below, we briefly summarize the role and relevance of each dimension:
> - Geometry Plausibility (GP) assesses the structural integrity and physical feasibility of the generated shape. **GP is a prerequisite to nearly all applications**—especially in physically grounded domains like robotics and simulation—where implausible geometry renders assets unusable.
> - Geometry Details (GD) reflects the fidelity of fine-scale structures, such as sharp edges and part boundaries. While GP ensures validity, GD distinguishes coarse models from high-quality assets. Therefore, **GD is essential for high-fidelity scenarios or immersive experiences**, such as AR/VR.
> - Texture Quality (TQ) evaluates the visual fidelity of surface textures in terms of resolution, realism, and aesthetic consistency. **TQ is especially important for visually-driven use cases** like gaming or film.
> - Geometry-Texture Coherency (GTC) assesses how well textures align with geometric features and part boundaries. While more subtle, **GTC offers crucial insight into 3D generation procedures**, where geometry and texture are synthesized jointly.
> - Prompt-3D Alignment (PA) evaluates the semantic and/or identity consistency with the input prompt. **PA is critical in conditional generation or editing tasks**, where output fidelity to the prompt is a core requirement.
>
> In sum, different applications prioritize different dimensions. GP and PA are often bottlenecks in failure cases (e.g., invalid geometry or prompt mismatch), whereas GD, TQ, and GTC enrich quality distinctions in successful outputs. A comprehensive evaluation across these dimensions supports both objective scoring and application-aware benchmarking.
>
> ### Q3: Render details
> Thank you for pointing this out. We acknowledge that rendering details such as tools and resolution can influence the perceived quality of visual outputs and potentially affect the evaluation outcomes. In our experiments, we **ensured consistent rendering settings across all methods and assets to minimize such bias**.
>
> Our rendering pipeline is implemented using both Blender and Kiui Python tools. Specifically, Blender is used for HDRI and point-light rendering due to its high-quality output, while Kiui is adopted for RGB and normal map rendering, benefiting from its computational efficiency and implementation simplicity. All videos are rendered at a default resolution of 512×512 pixels with 25 FPS, and we maintain consistent lighting and camera parameters across all methods. The average rendering cost per video is 17.47s using Blender.
>
> Given that all generation results are rendered under identical settings, any rendering-related effects are uniformly applied and thus do not influence the relative rankings in our evaluation. We will include these rendering details and provide relevant rendering scripts in the revised version to ensure completeness and reproducibility.
>
> ### Q4: Analysis on 15,000 assets
> Our benchmark includes a total of 15,300 synthesized 3D assets, generated by 30 generative methods from 510 prompts. We will analyze its diversity and representativeness from two perspectives: the generative methods involved and the composition of the prompt set.
> - **In terms of generative methods involved**, we include 9 text-to-3D generative models and 21 image-to-3D generative models. Among them contains recent **state-of-the-art approaches** (e.g., Hunyuan3D 2.0, MVDream) as well as **representative methods in different generative paradigms**. For instance, we cover SDS-based optimization methods (e.g., DreamFusion, Magic3D), multi-view synthesis approaches (e.g., MVDream, SyncDreamer), 3D reconstruction pipelines (e.g., LRM, GRM), and naive 3D generation models (e.g., TRELLIS, Hunyuan3D 2.0). Moreover, those models also spans **a wide range of 3D representations**, including Point cloud (e.g. Point-E), Mesh (e.g. Unique3D), NeRF (e.g. Latent-NeRF), 3D Gaussian Splatting (e.g. GRM), and so on. The complete list of models is provided below.
>     - **9 Text-to-3D generative models**: Mvdream, Lucid-dreamer, Magic3, GRM, Dreamfusion, Latent-NeRF, Shap-E, SJC, and Point-E.
>     - **21 Image-to-3D generative models**: Trellis, Hunyuan3D 2.0,  SPAR3D, InstantMesh, TripoSR, Unique3D, CRM, LN3Diff, Wonder3D, OpenLRM, Stable Zero123, Zero-1-to-3 XL, Magic123, LGM, GRM, SyncDreamer, Shap-E, Triplane-Gaussian, Point-E, EscherNet, and Free3D.
>
> - **In terms of the composition of the prompt set**, it was carefully curated to ensure diversity, representativeness and challenge. Specifically, the prompts span a wide range of object categories and vary in sentence lengths, object counts, spatial relations and attribute combinations, thereby introducing different levels of generation difficulty.
>
> While the dataset is synthesized, we acknowledge that there may be **potential discrepancies or biases when compared to real-world assets**. While this bias have limited impact on model evaluation—since all methods are assessed under the same synthesized setting—it could negatively affect the training of automated scorers. To mitigate this issue, we incorporated a subset of high-quality scanned 3D assets from Objaverse and OmniObject3D during the training phase, enhancing the realism and diversity of the data used for supervision.
>
> ### Q5: Reply for additional feedback
> - **About the release of inference weights.** Yes. We will release the inference weights before the camera-ready version to enable public evaluation. We will also provide an online leaderboard to support ongoing benchmarking and facilitate community.
> - **About the need of 3D-based scoring.** Yes, performing 3D-based scoring on the complete mesh can provide a more comprehensive assessment of overall geometric quality. While image-based evaluations (e.g., via multi-view renderings) are effective for capturing perceptual fidelity and visual coherence, they may overlook structural issues that are not visible from limited viewpoints, such as topology defects, inner holes, or self-intersections.

---

> > ### Comment · Reviewer_7bHR · 2025-08-05
> >
> > The rebuttal resolves my concerns, and I appreciate the detailed response. I will maintain the positive score.

---

> > > ### Author Response · Authors · 2025-08-05
> > >
> > > Thank you very much for your positive feedback and for taking the time to carefully consider our responses. We truly appreciate your support and encouragement throughout this process.

---

### Official Review · Reviewer_tpg3 · 2025-06-30

**Rating:** 5
**Confidence:** 4

**Summary:**

This paper systermatically constructs a 3D asset quality assessment workflow. The main efforts include curating a high-quality 3D asset dataset with high-quality labels produced by multiple agents. Notably, material evaluation is also involved. Moreover, a hybrid 3D-aware scoring framework is explored with video-based representations and part-level cues. The automated scoring framework is applied to different state-of-the-art 3D generative models to show their performances.

**Dataset Code Accessibility:**

Yes

**Ethical Considerations:**

No, there are no or only very minor ethics concerns

**Final Justification:**

Based on the authors' effective response, I decide to raise the score.

**Limitations Weaknesses:**

My major concern is about the actual alignment with human assessment. Sec 3.3 presents some descriptions and statistics about the conducted human-agent alignment study. However, the authors must provide more detailed information to make this study convincing, such as:
    -- How many human annotators are involved in the scoring? What are their profiles and backgrounds? What is the specific annotation process?
    -- In the top-right panel of Figure 2, why is there no specific numerical scale on the y-axis (indicating the L_1 loss)?
    -- Without strong demonstration of the human alignment level of the agent-produced scores, the subsequent experimental results and ablation studies will be meaningless.

It is introduced that Hi3DBench comprises 15,000 procedurally generated assets. According to the implementation details in Sec A.2 of the supplementary material, there are totally 8 generation models used, each of which produces 510 assets. No further information is provided.

**Strengths Contributions:**

Evaluating the quality of the generated 3D assets from multiple perspectives is a very important and practical task. This work has the following strengths:
1. Diversified 3D assets generated from multiple different generators. Part segmentation is also applied to enable subsequent part-level manipulations.
2. High-quality scores are obtained by designing hierarchical and comprehensive prompts and combining most advanced MLLMs,
3. Experiments and necessary ablation studies demonstrate the effectiveness of this work.

---

> ### Author Rebuttal · Authors · 2025-07-30
>
> Thank you for your thoughtful and constructive feedback. We appreciate your interest in the human-agent alignment aspect of our evaluation framework. Ensuring that the automatic scores meaningfully reflect human judgment is indeed a critical goal of our work. We address your concerns point by point below, providing additional details on annotator profiles and backgrounds, the specific annotation process, and the visualization in Figure 2.
>
> ### Q1: Information about the human annotations
> The human annotation data used in our thesis comes from two sources:
> - **Open-source annotations for object-level evaluation**.
> Under the same standardized criteria for object-level evaluation, we directly adopt human annotations from 3DGen-Bench [1].
>     - **Annotators.** According to its thesis, the 3DGen-Bench team employed **47 professional annotators** via a crowdsourcing platform. Among them, 12 are male and 35 are female. In terms of age distribution, 34% of annotators are between 18 and 24 years old, 34% are between 25 and 30, and 32% are over 30. Regarding education, 83% of the annotators have a college degree or higher, and 32% hold at least a bachelor's degree.
>     - **Annotation process.** To ensure annotation quality, they provided detailed guidelines, conducted regular monitoring, and proposed a **"Rank-and-Rate" protocol**: annotators first rank assets generated from the same prompt, then assign dimension-wise scores. Each asset is independently evaluated by two annotators, and multiple validation strategies are applied for data cleaning. Please refer to their thesis for full annotation details.
>     - **Statistics.** We sample 87 text prompts and 86 image prompts, yielding **1210 annotated assets**.
> - **In-house user study for part- and material- level evaluation**.
> Since existing benchmarks (e.g., 3DGen-Bench) focus only on object-level evaluation, we conducted a dedicated user study for part- and material-level assessment.
>     - **Annotators.** We recruited a total of **8 expert human annotators** (5 females and 3 males), all of whom are **Ph.D. students** with prior experience in 3D modeling or evaluation, ensuring a solid understanding of the assessment criteria.
>     - **Annotation process.** To promote scoring consistency, we designed a **detailed annotation protocol** (expanded from prompt templates illustrated in Figure S4 and S5), which includes explicit definitions and example visualizations for each evaluation dimension. All annotators also underwent a **calibration phase** before annotation. To reduce individual bias, each sample was independently rated by **at least 3 annotators**, and the final score is obtained by **averaging individual ratings**.
> In practice, the annotations are collected via **structured questionnaires**. For material-level annotations, we follow the **"Rank-and-Rate" strategy**, which encourages comparative judgment, helping annotators develop a more consistent internal scale and reducing scoring drift across samples.
>     - **Statistics.** We select 25 test prompts and sample 3-4 assets per prompt, resulting in **90 annotated assets**. For the part-level annotation, we sample 40 test assets and select 3-4 parts from each, yielding **159 annotated parts**.
>
> We will release those human annotation data and revise the manuscript to include detailed descriptions of the annotation process to enhance transparency and reproducibility.
>
> [1] Zhang Y, Zhang M, Wu T, et al. 3DGen-Bench: Comprehensive Benchmark Suite for 3D Generative Models[J]. arXiv preprint arXiv:2503.21745, 2025.
>
> ### Q2: Clarification on Figure 2 and human-agent alignment.
> Thank you for your careful observation. The top-right panel of Figure 2 is intended to highlight the relative trends in L₁ loss among different annotation methods, emphasizing comparative performance. To maintain visual clarity and avoid clutter, we intentionally omitted specific numerical scales on the y-axis. Meanwhile, for readers seeking exact values, the precise L₁ loss values are reported in **Table R1** in the Appendix. To improve clarity in the revised manuscript, we will consider adding numerical scales or annotations in Figure 2 to enhance interpretability.
>
> For reference, the precise values are listed below. Notably, widely-used agents such as GPT-4.1 and Claude-3.7 yield L1 losses of 0.838 and 1.100, respectively, whereas our full annotation pipeline (M²AP) achieves a substantially lower L1 loss of 0.257, clearly outperforming all baselines and ablation variants. This **strong alignment with human annotations** provides solid empirical support for the validity of our subsequent experiments and ablation analyses.
>
> | Method | GPT-4.1 | Claude-3.7 | Gemini-2.5 | Gork-3 | o3/o4 mini | M²AP (w/o Physical) | M²AP (w/o Reflection) | M²AP |
> |:------:|:------:|:------:|:------:|:------:|:------:|:------:|:------:|:------:|
> | **L1 loss** | 0.838 | 1.100 | 1.020 | 0.920 | 0.702 | 0.568 | 0.476 | **0.257** |
>
> ### Q3: More information for generated assets
> We sincerely apologize for the confusion caused by the incremental-styled presentation of our 3D asset statistics in Section A.2, which intends to highlight the contributions in data scale. To clarify, our benchmark includes a total of **15,300 synthesized 3D assets**, generated by **30 generative methods** from 510 prompts. The complete list of models is provided below.
> - **9 Text-to-3D generative models**: Mvdream, Lucid-dreamer, Magic3D, GRM, Dreamfusion, Latent-NeRF, Shap-E, SJC, and Point-E.
> - **21 Image-to-3D generative models**: Trellis, Hunyuan3D 2.0, SPAR3D, InstantMesh, TripoSR, Unique3D, CRM, LN3Diff, Wonder3D, OpenLRM, Stable Zero123, Zero-1-to-3 XL, Magic123, LGM, GRM, SyncDreamer, Shap-E, Triplane-Gaussian, Point-E, EscherNet, and Free3D.
>
> We hope this clarification helps contextualize the scale and comprehensiveness of our benchmark. And we will modify the content in Section A.2 in the revised version.

---

> > ### Author Response · Authors · 2025-08-05
> >
> > We hope our responses have effectively addressed your concerns regarding human-agent alignment. Your comments are valuable in improving the clarity of our work. Should there be any remaining questions or concerns, we would be grateful for the opportunity to provide further clarifications. We sincerely hope our explanations positively influence your final assessment.

---

> > ### Comment · Reviewer_tpg3 · 2025-08-07
> >
> > Thanks for the authors' detailed explanations. My major concerns and confusions have been basically addressed.

---

> > > ### Author Response · Authors · 2025-08-08
> > >
> > > Thank you again for your helpful feedback. We’re glad that our responses have addressed your main concerns. We hope the clarifications have helped improve your impression of the work, and we remain open to any suggestions that could help us further improve the work.

---

> ### Comment · Area_Chair_QvPA · 2025-08-07
> **Author-Reviewer discussion reminder**
>
> Dear Reviewer tpg3,
>
> Please read the rebuttal from the authors. You must participate in discussions with the authors before submitting the “Mandatory Acknowledgement” by Aug 8, 11:59 pm.
>
> Best regards,
>
> Area Chair

---

### Official Review · Reviewer_iTR4 · 2025-07-01

**Rating:** 5
**Confidence:** 4

**Summary:**

This paper introduces Hi3DEval, a novel hierarchical framework for evaluating the quality of 3D generative content. Recognizing the limitations of existing image-based and object-level metrics, Hi3DEval adopts a more comprehensive approach that combines both object-level and part-level assessments, and explicitly assesses material realism. The paper also introduces a large-scale annotated 3D dataset, and develops a 3D-aware automated scoring system using hybrid 3D representations. Experiments show improved alignment with human preferences and superior performance over existing metrics.

**Dataset Code Accessibility:**

Yes

**Ethical Considerations:**

No, there are no or only very minor ethics concerns

**Final Justification:**

The rebuttal addresses most of my concerns. I will maintain my current score.

**Limitations Weaknesses:**

1. This paper employs PartField for part-level segmentation of generated 3D assets. However, PartField disentangles field features through unsupervised clustering, often yielding unreasonable segmentation results with severe aliasing artifacts at boundary regions. I am concerned this may compromise annotation and evaluation reliability.
2. The paper evaluates various 3D generation methods using 510 prompts. Given that works like Trellis [1] and Clay [2] utilize 3.2k and 16k prompts respectively, I question whether this sample size is statistically sufficient.

[1] Xiang J, Lv Z, Xu S, et al. Structured 3d latents for scalable and versatile 3d generation. CVPR 2025.
[2] Zhang, Longwen, et al. Clay: a controllable large-scale generative model for creating high-quality 3d assets. TOG 2024.

**Strengths Contributions:**

1. This paper is well-written, with both the motivation and framework clearly articulated.
2. This paper propose a comprehensive evaluation framework for 3D generative models, enabling object-level, part-level, and material-level assessment of generated 3D assets.

---

> ### Author Rebuttal · Authors · 2025-07-30
>
> We sincerely thank you for your thoughtful and constructive feedback. We're encouraged by your recognition of the significance and potential of our work. In the following, we provide point-by-point responses to your comments.
>
> ### Q1: Employment of PartField for part-level segmentation.
> We acknowledge that PartField, as an unsupervised method, can produce imperfect segmentation results. However, we chose PartField primarily for its category-agnostic generalization ability and scalability across large-scale, diverse 3D assets. Compared to existing methods such as SAMPart3D and SAMesh, **PartField has achieved SOTA performances** in terms of both segmentation quality and computational cost. More details on this comparison are provided in Section A.2.
>
> Additionally, in the context of our 3D quality evaluation task, **the primary goal of part-level segmentation** is not to recover precise semantic parts or enforce strict semantic boundaries, but rather to localize perceptually meaningful regions for fine-grained quality assessment. Therefore, **a certain degree of imperfection or noise is acceptable**. As illustrated in Figure 7, despite minor boundary artifacts—such as around the giraffe's front thigh or the cat's head—the segmentation results remain generally sufficient for identifying flawed regions in the generated assets.
>
> Finally, we agree that improving segmentation quality remains a valuable direction for future refinement. Since part segmentation serves as a preprocessing step in our pipeline, our framework is compatible with future improvements—and **a more reliable method can be easily substituted when available**.
>
> ### Q2: Statistical sufficiency of prompt scale.
> Thank you for raising this important point. We agree that the number of prompts is a critical factor in ensuring a statistically reliable benchmark. Our current benchmark is built on 510 carefully curated prompts, which strike a **balance between diversity, difficulty, representativeness, and annotation feasibility**. Despite its moderate scale, the prompt set already **reveals consistent and meaningful performance differences across methods**.
>
> Specifically, the prompt set spans **a wide range of object categories** and **varies in sentence lengths, object counts, spatial relations, and attribute combinations**. This variety introduces different levels of generation difficulty and allows for robust benchmarking across multiple axes. Specifically, the textual prompts cover over 270 distinct categories derived from 6 basic subjects: “Vehicle”, “Animal”, “Plant”, “Food”, “Indoor objects”, “Outdoor objects”. In terms of sentence length, there are 83 prompts containing <=5 words, 230 containing 6-10 words, 109 containing 11-15 words, and 88 containing >15 words. In terms of semantic attributes, there are exactly 43 prompts with explicit counts (e.g., "Three vibrant balloons tied together"), 104 with implicit counts (e.g., “A herd of migrating elephants”), 73 featuring spatial relationships (e.g., "A book left on a park bench"), and 290 individual objects (e.g., "A wooden bicycle").
>
> Finally, our benchmark is designed with scalability in mind. As the field evolves, we **remain open to expanding the prompt set in future updates** if broader coverage is empirically shown to benefit the evaluation process.

---

> > ### Comment · Reviewer_iTR4 · 2025-08-04
> >
> > Thank you for the detailed responses, which address most of my concerns. I will maintain my current score.

---

> > > ### Author Response · Authors · 2025-08-04
> > >
> > > Thank you sincerely for your kind follow-up and for taking the time to review our responses. We deeply appreciate your constructive feedback throughout the process and your continued support of our work. Your insights have been invaluable in helping us enhance the clarity and quality of the paper.

---

### Official Review · Reviewer_LMGB · 2025-07-01

**Rating:** 4
**Confidence:** 4

**Summary:**

Hi3DEval is a hierarchical evaluation framework accompanied by a dataset (Hi3DBench) and a hybrid automated scoring system, designed to comprehensively evaluate 3D generative objects, achieving better human alignment than existing methods and benchmarks.

**Dataset Code Accessibility:**

Yes

**Ethical Considerations:**

No, there are no or only very minor ethics concerns

**Final Justification:**

Thank the author for the response. Most of my concerns have been addressed. I hope the authors will follow through on their commitment to organize the codebase and provide an easy-to-run scoring pipeline.

**Limitations Weaknesses:**

Despite its strengths, several aspects of this paper lack sufficient clarity and detail, which impacts its overall contribution and utility:

1. **Human Annotation Clarity**: The calibration process for the human data used in quantitative measurements is not sufficiently described, impacting reproducibility.

2. **Evaluation Resource Requirements**: The manuscript lacks details on the computational resources or VLM API costs required for running the evaluations, which is crucial for practical adoption.

3. **Automated Benchmark Workflow**: The provided code does not clearly automate the entire benchmark evaluation process, making the practical application for future research challenging.

4. **Scoring Model Discussion**: The paper proposes video-based and 3D-based score models to automatically evaluate the quality of 3D object generation, which is good and more convenient than complex calls to multiple large-scale VLM APIs. However, the experimental section does not discuss these scoring models in detail. Were the quantitative data in the experiments obtained by the scoring models or by M^2AP?

5. **Scoring Model Availability**: The open-source code and data do not include the scoring models, further reducing the practicality of the benchmark proposed in the paper.

**Strengths Contributions:**

1. Hi3DBench provides a more accurate benchmark for comparing different generation techniques, which also includes part-level and physical material evaluation, in addition to object-level assessment. It can guide future model improvements towards higher fidelity and realism.

2. Previous benchmarks for 3D object generation are well discussed, and this work explicitly highlights the shortcomings of previous methods.

3. The manuscript is well-written and easy to read.

---

> ### Author Rebuttal · Authors · 2025-07-30
>
> We sincerely thank you for your valuable comments and thoughtful questions. We deeply appreciate the time and effort devoted to evaluating our work. In the following, we provide detailed responses to each concern, clarifying key aspects of our contributions and addressing potential confusions.
>
> ### Q1: Human Annotation Clarity
> Thank you for your question regarding the calibration process. We will provide the complete annotation data along with detailed descriptions of the annotation procedure in the revised version. To clarify, the human evaluation data used in our quantitative analysis comes from two sources:
>
> - **Open-source annotations for object-level evaluation**.
> Under the same standardized criteria for object-level evaluation, we directly adopt human annotations from 3DGen-Bench [1].
>     - According to its thesis, the 3DGen-Bench team employed **47 professional annotators** via a crowdsourcing platform.
>     - To ensure annotation quality, they provided detailed guidelines, conducted regular monitoring, and proposed a **"Rank-and-Rate" protocol**: annotators first rank assets generated from the same prompt, then assign dimension-wise scores. Each asset is independently evaluated by two annotators, and multiple validation strategies are applied for data cleaning. Please refer to their thesis for full annotation details.
>     - Finally, we sample 87 text prompts and 86 image prompts, yielding **1210 annotated assets**.
> - **In-house user-study for part- and material- level evaluation**.
> Since existing benchmarks (e.g., 3DGen-Bench) focus only on object-level evaluation, we conducted a dedicated user study for part- and material-level assessment.
>     - We recruited a total of **8 expert human annotators** (5 females and 3 males), all of whom are **Ph.D. students** with prior experience in 3D modeling or evaluation, ensuring a solid understanding of the assessment criteria.
>     - To promote scoring consistency, we designed a **detailed annotation protocol** (expanded from prompt templates illustrated in Figure S4 and S5), which includes explicit definitions and example visualizations for each evaluation dimension. All annotators also underwent a **calibration phase** before annotation. To reduce individual bias, each sample was independently rated by **at least 3 annotators**, and the final score is obtained by **averaging individual ratings**. In practice, the annotations are collected via **structured questionnaires**. For material-level annotations, we follow the **"Rank-and-Rate" strategy**, which encourages comparative judgment, helping annotators develop a more consistent internal scale and reducing scoring drift across samples.
>     - We select 25 test prompts and sample 3-4 assets per prompt, resulting in **86 annotated assets**. For the part-level annotation, we sample 40 test assets and select 3-4 parts from each, yielding **159 annotated parts**.
>
> [1] Zhang Y, Zhang M, Wu T, et al. 3DGen-Bench: Comprehensive Benchmark Suite for 3D Generative Models[J]. arXiv preprint arXiv:2503.21745, 2025.
>
> ### Q2: Evaluation Resource Requirements
> The annotation process using Vision-Language Models (VLMs) in our Multi-agent Multi-modal Annotation Pipeline (M²AP) is highly efficient. Without parallelization, **annotating a single object typically takes 20 to 60 seconds**, primarily influenced by network latency. In terms of cost, completing one full M²AP annotation—which involves multiple VLM API calls—costs **approximately $0.15 per object**. Detailed cost statistics for each configuration are presented in the table below. We will also explicitly include these details in the revised manuscript to ensure full transparency.
>
> | Annotation type | Object-level | Part-level | Material-level |   Total |
> |:----------|:----------:|:----------:|:----------:|:----------:|
> | **Cost (USD)** | 2.5k | 1.0k | 0.6k | 4.1k |
>
> ### Q3: Automated Benchmark Workflow
> Thank you very much for your valuable feedback. Due to the policies outlined by the conference committee, we are currently unable to update the code repository during the rebuttal period. While the **essential components—including the scoring model and streamlined evaluation scripts—are already included in the repository**, we agree that the workflow could benefit from clearer organization and documentation. And we will **promptly update the GitHub repository to include well-documented instructions and the official pretrained checkpoints** used in our experiments to ensure ease of use and reproducibility.
>
> The current codebase is organized as follows:
> - The `configs/directory` contains necessary configuration files;
> - The `dataset/directory` includes scripts for constructing training and inference datasets;
> - The `models/directory` implements the core model architecture;
> - The `utils/directory` contains auxiliary utility functions;
> - The `infer.py` script serves as the main entry point for inference and scoring, integrating components from the directories mentioned above;
> - The `config.py` file specifies paths to datasets and checkpoints, which users can modify as needed;
> - Finally, the `run.sh` script provides a simple command-line interface to execute the full scoring process.
>
> ### Q4: Scoring Model Discussion
> To clarify, all the quantitative results reported in the experimental sections (including Figure 5, Table 1, Table 2, Table R2, Table R4, Table R5, Table R6, and Table R7) are obtained using these trained **scoring models**. The M²AP protocol is used exclusively during data curation, serving to generate consistent and reliable supervision signals for training and validation, rather than for evaluation itself.
>
> ### Q5: Scoring Model Availability
> We appreciate your feedback. We understand that the absence of detailed documentation may have caused confusion. However, the scoring model is already included in our released repository. Specifically, the scoring head can be found at `configs/head/modeling_head.py`. Users can modify the configuration in `config.py` to specify the data and checkpoint paths and then simply run the `run.sh` script in the root directory to perform end-to-end evaluation.
>
> We will update the repository with comprehensive documentation and pretrained checkpoints to ensure clarity and ease of use once the conference committee lifts the GitHub modification restriction.

---

> > ### Author Response · Authors · 2025-08-05
> >
> > We would greatly appreciate it if you could let us know whether our rebuttal has addressed your concerns. We remain happy to follow up on any further questions or issues you may have. Your insights are invaluable to us, and we sincerely hope our clarifications will positively inform your final assessment.

---

> > ### Comment · Reviewer_LMGB · 2025-08-05
> >
> > Thank you for the response. Most of my concerns have been addressed. I hope the authors will follow through on their commitment to organize the codebase and provide an easy-to-run scoring pipeline. I will update my rating to * Borderline Accept*, provided that no new major issues arise during the review-AC discussion phase.

---

> > ### Author Response · Authors · 2025-08-05
> >
> > Thank you very much for your thoughtful feedback, and we are glad to hear that most of your concerns have been addressed. We would like to reassure you that we will make it a priority to reorganize the codebase along with an easy-to-run scoring pipeline, as promised. Your constructive suggestions have been very helpful, and we sincerely thank you for your consideration.

---

### Note · Authors · 2025-08-16

We sincerely thank all reviewers and the area chair for their time, constructive feedback, and positive recognition of our work. The encouraging comments and insightful suggestions have been instrumental in improving the quality and clarity of our paper.

**We are pleased that the novelty, effectiveness, and main contributions of Hi3DEval have been well recognized.** Specifically, reviewer LMGB highlighted our benchmark as “a more accurate benchmark”, while reviewer iTR4 described it as “a comprehensive evaluation framework for 3D generative models, enabling object-level, part-level, and material-level assessment”. Meanwhile, reviewer tpg3 acknowledged our work as “a very important and practical task,” further noting that “experiments and necessary ablation studies demonstrate the effectiveness of this work”. In addition, reviewer 7bHR agreed that our method “consistently outperforms strong baselines”.

In response to the reviewers’ comments, we provided additional details regarding the rendering process (for 7bHR), annotator profiles and the annotation process (for LMGB and tpg3), as well as supplementary discussions on 3D part segmentation (for iTR4 and 7bHR), human-agent alignment (for tpg3), and the availability of the scoring model (for LMGB). **We are glad that all reviewers confirmed their major concerns have been addressed.**

To enhance transparency and reproducibility, we will incorporate these clarifications in the final version and update the code repository to be more well-organized. **We believe Hi3DEval fills a critical gap in the community** by providing the first scalable and fine-grained benchmark for 3D generative models, which will serve as a foundation for fair comparison and future progress in the rapidly growing 3D generation field.

Once again, we sincerely thank the reviewers and the area chair for their thoughtful engagement throughout the review process. And we would be truly grateful if the area chair could kindly take our clarifications and the significance of Hi3DEval into thoughtful consideration.

---

### Decision · Program_Chairs · 2025-09-18

**Decision:**

Accept (poster)

**Comment:**

This paper introduces Hi3DEval, a hierarchical evaluation framework for 3D generative models, along with the associated Hi3DBench dataset and a suite of hybrid automated scoring models. The reviewers unanimously acknowledge the technical strengths, practical utility, and positive impact of this work on the evaluation of 3D generative content. The paper is recommended for acceptance, with the expectation that the authors will implement their promised improvements prior to publication.